# G-Retriever: Retrieval-Augmented Generation for Textual Graph Understanding and Question Answering

**Xiaoxin He**[1]   **Yijun Tian**[2]   **Yifei Sun**[1]   **Nitesh V. Chawla**[2]   **Thomas Laurent**[3]

**Yann LeCun**[4,5]   **Xavier Bresson**[1]   **Bryan Hooi**[1]

{xiaoxin, yifeisun, xaviercs, bhooi}@comp.nus.edu.sg
{yijun.tian, nchawla}@nd.edu, tlaurent@lmu.edu, yann@cs.nyu.edu

[1]National University of Singapore   [2]University of Notre Dame   [3]Loyola Marymount University
[4]New York University   [5]Meta AI

## Abstract

Given a graph with textual attributes, we enable users to 'chat with their graph': that is, to ask questions about the graph using a conversational interface. In response to a user's questions, our method provides textual replies and highlights the relevant parts of the graph. While existing works integrate large language models (LLMs) and graph neural networks (GNNs) in various ways, they mostly focus on either conventional graph tasks (such as node, edge, and graph classification), or on answering simple graph queries on small or synthetic graphs. In contrast, we develop a flexible question-answering framework targeting real-world textual graphs, applicable to multiple applications including scene graph understanding, common sense reasoning, and knowledge graph reasoning. Toward this goal, we first develop a Graph Question Answering (GraphQA) benchmark with data collected from different tasks. Then, we propose our *G-Retriever* method, introducing the first retrieval-augmented generation (RAG) approach for general textual graphs, which can be fine-tuned to enhance graph understanding via soft prompting. To resist hallucination and to allow for textual graphs that greatly exceed the LLM's context window size, *G-Retriever* performs RAG over a graph by formulating this task as a Prize-Collecting Steiner Tree optimization problem. Empirical evaluations show that our method outperforms baselines on textual graph tasks from multiple domains, scales well with larger graph sizes, and mitigates hallucination. Our codes and datasets are available at: `https://github.com/XiaoxinHe/G-Retriever`.

## 1   Introduction

**Graphs and Large Language Models (LLMs).** The advent of LLMs has significantly shaped the artificial intelligence landscape. As these models are applied to increasingly diverse tasks, their ability to process complex structured data will be increasingly vital. In particular, in our interconnected world, a significant portion of real-world data inherently possesses a graph structure, such as the Web, e-commerce, recommendation systems, knowledge graphs, and many others. Moreover, many of these involve graphs with textual attributes (*i.e., textual graphs*), making them well-suited for LLM-centric methods. This has spurred interest in combining graph-based technologies, particularly graph neural networks (GNNs), with LLMs to enhance their reasoning on graphs [44, 15, 24].

**The Present Work: Enabling 'Chat With Your Graph'.** While existing works integrate LLMs and GNNs in various ways, they mostly focus on conventional graph tasks such as node, edge and graph

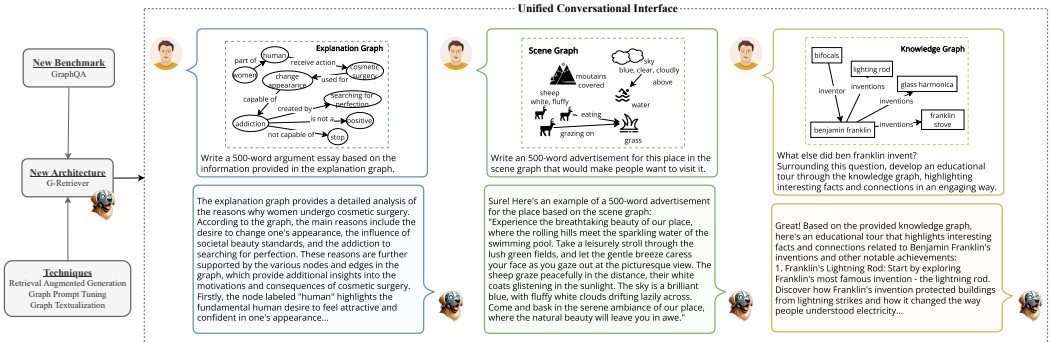

Figure 1: We develop a flexible question-answering framework targeting real-world textual graph applications via a unified conversational interface. Presented here are examples showcasing the model's adeptness in handling generative and creative queries in practical graph-related tasks: common sense reasoning, scene understanding, and knowledge graph reasoning, respectively.

classification [8], or answering simple questions on small or synthetic graphs [44, 31]. In contrast, we develop a flexible question-answering framework targeting complex and real-world graphs. This framework enables users to 'chat with their graph' via a unified conversational interface, representing a leap towards intuitive interaction with graph data, as demonstrated in Figure 1.

**The Need for a Comprehensive GraphQA Benchmark.** Question answering (QA) is a fundamentally important task in natural language processing, serving as a key benchmark for assessing LLMs and providing a unified interface for various capabilities. Despite extensive research in QA, a comprehensive benchmark specifically tailored for the graph modality is lacking. In contrast to existing benchmarks that focus on basic graph-based reasoning tasks such as node degree, edge existence, and shortest path [6, 44], our benchmark addresses complex and real-world graph applications including common sense reasoning, scene understanding, and knowledge graph reasoning (refer to Figure 2). This is vital for measuring progress toward a model capable of answering a wide range of questions about graphs from diverse applications.

**New Architecture for GraphQA.** To enable effective and efficient graph QA, even on large graphs, we propose *G-Retriever*, a new framework combining the strengths of GNNs, LLMs, and RAG (Figure 3). Next, we will discuss the motivation, strengths, and details of our model.

**Tackling Hallucination in Graph LLMs.** LLMs are prone to hallucination, a phenomenon where the generated content is factually inaccurate or nonsensical [12]. We validate the presence of this issue in graph settings. In particular, we employ a baseline method that adapts MiniGPT-4 [57] to graphs, where a frozen LLM interacts with a trainable GNN that encodes graph data as a soft prompt, as in GraphToken [31]. Our findings, shown in Table 1, indicate that hallucination, an important problem in text-based LLMs, is also prevalent in Graph LLMs. This may be attributed to the baseline's inability to recall the entire graph structure from a single graph embedding, leading to the generation of incorrect nodes or edges during the QA task. In contrast, by employing RAG for direct information retrieval from the actual graph, our *G-Retriever* mitigates this issue, as substantiated by Table 1.

**Enhancing Scalability and Efficiency in Graph LLMs.** Recent research endeavors have explored translating graphs into natural language, such as by flattening nodes and edges into a text sequence, enabling their processing by LLMs for graph-based tasks [56, 6]. However, this method faces critical scalability issues. Converting a graph with thousands of nodes and edges into a text sequence results in an excessive number of tokens, surpassing the input capacity of many LLMs. An alternative of truncating the graph text sequence to fit the LLM's input token limit leads to loss of information and response quality. *G-Retriever* overcomes these issues with its RAG component, which allows for effective scaling to larger graphs by selectively retrieving only relevant parts of the graph.

**Tailoring the RAG Approach to Graphs.** Existing RAG methodologies are primarily designed for simpler data types or knowledge graphs, where information is retrieved in a manner isolated from the graph structure [7, 1, 36, 16]. Hence, we introduce a new retrieval approach for general textual graphs. Notably, we formulate subgraph retrieval as a Prize-Collecting Steiner Tree (PCST)

Table 1: Observation and mitigation of hallucination in graph LLMs.

| | |
|---|---|
| User | (graph diagram) elephants —in→ bushes; elephants —eating→ grass; elephant —standing by→ river; bushes; grass —on the side of→ river 

 What kind of animal is in the bushes? 
 Answer the question and list the nodes or edges from the graph that support your answer. |
| LLM w/ Graph Prompt Tuning | [Incorrect response with hallucinated nodes and edges] 
 The animal in the bushes is a deer. 
 Nodes: 
 * Deer (node 1) 
 * Bushes (node 2) 
 Edges: 
 * Deer → Bushes (edge 1) 
 * Deer → Grass (edge 2) 
 * Bushes → Grass (edge 3) |
| *G-Retriever* (Ours) | [Correct response with accurate node and edge references] 
 Based on the information provided in the graph, the animal in the bushes is an elephant. 
 The nodes that support this answer are: 
 * Node 1: The elephant is mentioned in the first line of the graph. 
 The edges that support this answer are: 
 * The edge connects the elephant to the bushes, indicating that the elephant is in the bushes. 
 Therefore, the answer to the question is: The animal in the bushes is an elephant. |

optimization problem, which takes the neighborhood information into account during retrieval. This also allows the return of a subgraph most relevant to a query, thereby improving explainability.

The contributions of this paper are outlined as follows:

- **Pioneering the integration of Graph RAG.** We present the first retrieval approach for general textual graph tasks, which greatly enhances scalability and efficiency.
- **Enabling 'Chat with Your Graph'.** We develop a flexible question answering framework to handle complex and real-world textual graphs through a unified conversational interface.
- **Introducing A Novel GraphQA Benchmark.** We introduce a diverse benchmark targeted at real-world graph question answering, filling a crucial research gap.
- **Empirical Findings.** We demonstrate the efficiency and effectiveness of *G-Retriever* in multiple domains and present the significant finding of hallucination in graph LLMs.

## 2 Related Work

**Graphs and Large Language Models.** A significant body of research has emerged at the intersection of graph-based techniques and LLMs [30, 24, 15, 44, 54]. This exploration spans diverse aspects, ranging from the design of general graph models [47, 25, 51, 19, 40, 31], and multi-modal architectures [23, 49] to practical applications. Noteworthy applications include fundamental graph reasoning [52, 3, 56], node classification [8, 11, 39, 5, 50, 4, 33], graph classification/regression [32, 55], and leveraging LLMs for knowledge graph-related tasks [41, 14, 29].

**Retrieval-Augmented Generation (RAG).** The concept of Retrieval-Augmented Generation, initially proposed by Lewis et al. [21], has gained increased attention for its ability to mitigate the issue of hallucination within LLMs and enhance trustworthiness and explainability [7]. Despite its success in language-related tasks, the application of retrieval-based approaches to general graph tasks remains largely unexplored. Most existing work focuses primarily on the knowledge graph [38, 1, 36, 16]. Our research is the first to apply a retrieval-based approach to general graph tasks, marking a novel advancement in the field and demonstrating the versatility of RAG beyond language processing.

**Parameter-Efficient Fine-Tuning (PEFT).** The field of LLMs has witnessed significant advancements through various parameter-efficient fine-tuning techniques. These methodologies have played a crucial role in refining LLMs, boosting their performance while minimizing the need for extensive parameter training. Notable among these techniques are prompt tuning, as introduced by Lester et al. [20], and prefix tuning, proposed by Li and Liang [22]. Furthermore, methods like LoRA [10],

and the LLaMA-adapter [53], have been influential. These advancements in PEFT have laid the foundation for the development of sophisticated multimodal models. Prominent examples in this domain include MiniGPT-4 [57], LLaVA [26], and NExT-Chat [46]. There are also emerging efforts in applying PEFT to graph LLMs, such as GraphLLM [3] and GraphToken [31] for basic graph reasoing tasks and GNP [41] for multi-option QA on knowledge graphs.

## 3 Formalization

This section establishes the notation and formalizes key concepts related to textual graphs, language models for text encoding, and large language models and prompt tuning.

**Textual Graphs.** A textual graph is a graph where nodes and edges possess textual attributes. Formally, it can be defined as $G = (V, E, \{x_n\}_{n \in V}, \{x_e\}_{e \in E})$, where $V$ and $E$ represent the sets of nodes and edges, respectively. Additionally, $x_n \in D^{L_n}$ and $x_e \in D^{L_e}$ denote sequential text associate with a node $n \in V$ or an edge $e \in E$, where $D$ represents the vocabulary, and $L_n$ and $L_e$ signify the length of the text associated with the respective node or edge.

**Language Models for Text Encoding.** In the context of textual graphs, language models (LMs) are essential for encoding the text attributes associated with nodes and edges, thereby learning representations that capture their semantic meaning. For a node $n$ with text attributes $x_n \in D^{L_n}$, an LM encodes these attributes as:

$$z_n = \text{LM}(x_n) \in \mathbb{R}^d, \tag{1}$$

where $z_n$ is the output of the LM, and $d$ is the dimension of the output vector.

**Large Language Models and Prompt Tuning.** LLMs have introduced a new paradigm for task-adaptation known as "pre-train, prompt, and predict", replacing the traditional "pre-train, fine-tune" paradigm. In this paradigm, the LLM is first pre-trained on a large corpus of text data to learn general language representations. Then, rather than fine-tuning the model on task-specific labeled data, the model is prompted with a textual prompt that specifies the task and context. Subsequently, the model generates the output directly based on the prompt and the input.

The LLM, parameterized by weights $\theta$, takes a sequence of tokens $X$, and a prompt $P$ as input, and generates a sequence of tokens $Y = \{y_1, y_2, \ldots, y_r\}$ as output. Formally, the probability distribution of the output sequence given the concatenated input sequence and prompt, *i.e.,* $[P; X]$, is defined as

$$p_\theta(Y|[P; X]) = \prod_{i=1}^{r} p_\theta(y_i|y_{<i}, [P; X]). \tag{2}$$

Here, $y_{<i}$ represents the prefix of sequence $y$ up to position $i - 1$, and $p(y_i|y_{<i}, [P; X])$ represents the probability of generating token $y_i$ given $y_{<i}$ and $[P; X]$.

Soft prompt tuning eliminates the need for manual prompt design. Given a series of $p$ tokens $X = \{x_1, x_2, \ldots, x_p\}$, after being processed by the text embedder, it forms a matrix $X_e \in \mathbb{R}^{p \times d_l}$, where $d_l$ is the dimension of the embedding space. Soft prompts can be represented as parameters $P_e \in \mathbb{R}^{q \times d_l}$, where $q$ is the length of the prompt. The prompt is then concatenated with the embedded input, forming a single matrix $[P_e; X_e] \in \mathbb{R}^{(q+p) \times d_l}$. This combined matrix is processed by the self-attention layers in LLM as usual. Training involves maximizing the likelihood of $Y$ through backpropagation, with gradient updates applied solely to $P_e$, while $\theta$ remains fixed.

## 4 Proposed GraphQA Benchmark

Our GraphQA represents a comprehensive and diverse benchmark for graph question-answering. It is tailored to assess the capabilities of models in answering a wide range of questions about graphs across diverse domains.

### 4.1 Data Format

Each entry in the GraphQA benchmark consists of a textual graph, a question related to the graph, and one or more corresponding answers, as illustrated in Figure 2.

Table 2: Summary of datasets used in GraphQA benchmark.

| Dataset | ExplaGraphs | SceneGraphs | WebQSP |
|---|---|---|---|
| #Graphs | 2,766 | 100,000 | 4,737 |
| Avg. #Nodes | 5.17 | 19.13 | 1370.89 |
| Avg. #Edges | 4.25 | 68.44 | 4252.37 |
| Node Attribute | Commonsense concepts | Object attributes (*e.g.,* color, shape) | Entities in Freebase |
| Edge Attribute | Commonsense relations | Relations (*e.g.,* actions, spatial relations) | Relations in Freebase |
| Task | Common sense reasoning | Scene graph question answering | Knowledge based question answering |
| Evaluation Matrix | Accuracy | Accuracy | Hit@1 |

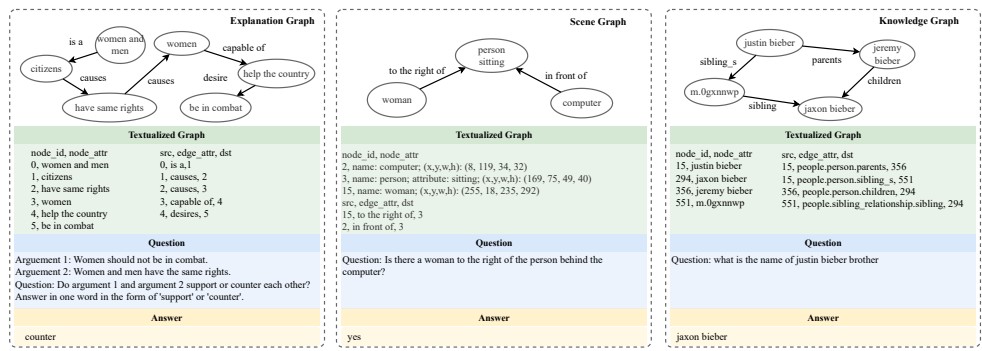

Figure 2: Illustrative examples from the GraphQA benchmark datasets.

**Textual Graphs.** The textual graph is converted into a natural language format, resulting in a list of nodes and edges, akin to a CSV file format. It is important to note that while multiple methods exist for textualizing a graph, our focus is not on identifying the optimal solution. Instead, we prioritize a straightforward yet empirically effective approach for representing graphs in natural language, facilitating the benchmark's use in diverse GraphQA scenarios.

**Questions and Answers.** Questions are designed to explore specific elements or relationships within the graph. Answers, residing within the attributes of nodes or edges, often require multi-hop reasoning for accurate identification.

## 4.2 Description of Datasets

The GraphQA benchmark integrates three existing datasets: ExplaGraphs, SceneGraphs, and WebQSP. Table 2 presents the summary statistics of these datasets. It is important to note that these datasets were not originally developed for this work. However, a significant contribution of our research is the standardization and processing of these diverse datasets into a uniform data format suitable for the GraphQA benchmark. These datasets, previously utilized in different contexts, are reintroduced with a new focus tailored for GraphQA. For a detailed comparison with the original datasets, see the Appendix C.

ExplaGraphs is a dataset for generative commonsense reasoning, focusing on creating explanation graphs for stance prediction in debates. It offers detailed, unambiguous commonsense-augmented graphs to evaluate arguments supporting or refuting a belief. The primary task is to assess whether arguments are supportive or contradictory, using accuracy as the metric. We have converted the triplet-form provided in Saha et al. [35] into a standard graph format.

SceneGraphs, a visual question answering dataset, includes 100,000 scene graphs. Each graph details objects, attributes, and relations within an image. This dataset challenges users with tasks requiring spatial understanding and multi-step inference. The task is to answer open-ended questions based on a textual description of a scene graph, evaluated on accuracy. We have sampled from the GQA dataset [13] and constructed standard graphs from the provided JSON files.

WebQSP is a large-scale multi-hop knowledge graph QA dataset consisting of 4,737 questions. It was proposed by Yih et al. [48] and, following Luo et al. [28], utilizes a subset of Freebase, encompassing facts within 2-hops of entities mentioned in the questions. The task involves answering questions that

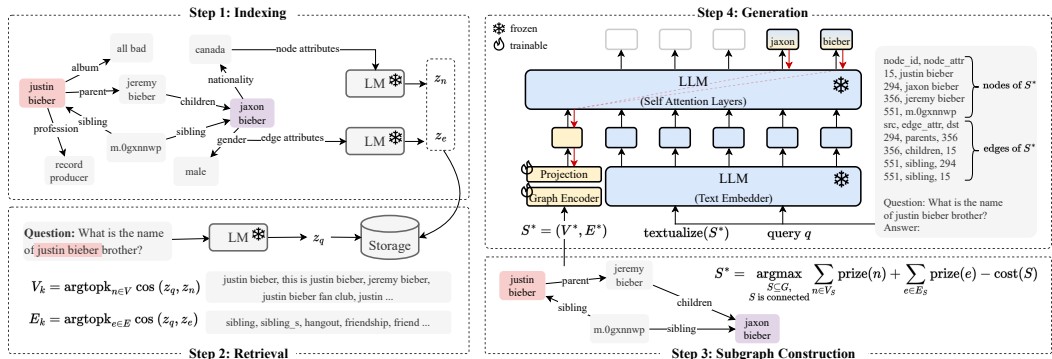

Figure 3: Overview of the proposed *G-Retriever*: 1) Indexing: Graphs are indexed for efficient query processing; 2) Retrieval: The most semantically relevant nodes and edges are retrieved, conditioned on the query; 3) Subgraph Construction: A connected subgraph is extracted, covering as many relevant nodes and edges as possible while maintaining a manageable graph size; 4) Generation: An answer is generated using a 'graph prompt', a textualized graph, and the query.

require multi-hop reasoning. Given the possibility of multiple answers for the same question, the hit@1 metric is used to assess the precision of the top returned answer.

## 5 G-Retriever

In this section, we introduce *G-Retriever*, a new architecture tailored for GraphQA, which integrates the strengths of GNNs, LLMs, and RAG. To allow efficient fine-tuning while preserving the LLM's pretrained language capabilities, we freeze the LLM and use a soft prompting approach on the output of the GNN. Our RAG-based design mitigates hallucinations through direct retrieval of the graph, while allowing our approach to scale to graphs exceeding the LLM's context window size. To adapt RAG to graphs, we formulate subgraph retrieval as a PCST optimization problem. This approach also allows us to enhance explainability by returning the retrieved subgraph.

*G-Retriever* comprises four main steps: indexing, retrieval, subgraph construction and generation, as depicted in Figure 3. The implementation details of each step are elaborated in the following sections.

### 5.1 Indexing

We initiate the RAG approach by generating node and graph embeddings using a pre-trained LM. These embeddings are then stored in a nearest neighbor data structure.

To elaborate, consider $x_n \in D^{L_n}$ as the text attributes of node $n$. Utilizing a pre-trained LM, such as SentenceBert [34], we apply the LM to $x_n$, yielding the representation $z_n$:

$$z_n = \text{LM}(x_n) \in \mathbb{R}^d, \tag{3}$$

where $d$ denotes the dimension of the output vector. Similar preprocessing steps are applied to edges. Refer to Figure 3, Step 1 for an illustrative representation.

### 5.2 Retrieval

For retrieval, we employ the same encoding strategy to the query $x_q$, to ensure consistent treatment of textual information:

$$z_q = \text{LM}(x_q) \in \mathbb{R}^d. \tag{4}$$

Next, to identify the most relevant nodes and edges for the current query, we use a k-nearest neighbors retrieval approach. This method yields a set of 'relevant nodes/edges' based on the similarity between the query and each node or edge. The retrieval operation is defined as:

$$V_k = \text{argtopk}_{n \in V} \cos(z_q, z_n)$$
$$E_k = \text{argtopk}_{e \in E} \cos(z_q, z_e), \tag{5}$$

where $z_n$ and $z_e$ are the embeddings of node $n$ and edge $e$, respectively. We use the cosine similarity function, $\cos(\cdot, \cdot)$, to measure the similarity between the query representation and the node/edge embeddings. The argtopk operation retrieves the top-k elements based on this similarity, providing a set of nodes $V_k$ and edges $E_k$ considered most relevant to the query. See Step 2 of Figure 3.

## 5.3 Subgraph Construction

This step aims to construct a subgraph that encompasses as many relevant nodes and edges as possible, while keeping the graph size manageable. This approach offers two key benefits: Firstly, it helps to filter out nodes and edges that are not pertinent to the query. This is crucial because irrelevant information can overshadow the useful data, potentially diverting the focus of the subsequent LLM from the information of interest. Secondly, it enhances efficiency; by keeping the graph size manageable, it becomes feasible to translate the graph into natural language and then input it into the LLM for processing. The Prize-Collecting Steiner Tree algorithm [2] serves as our primary method for identifying such optimally sized and relevant subgraphs. See Step 3 in Figure 3.

**Prize-Collecting Steiner Tree (PCST).** The PCST problem aims to find a connected subgraph that maximizes the total prize values of its nodes while minimizing the total costs of its edges. Our approach assigns higher prize values to nodes and edges more relevant to the query, as measured by cosine similarity. Specifically, the top $k$ nodes/edges are assigned descending prize values from $k$ down to 1, with the rest assigned zero. The node prize assignment is as follows:

$$\text{prize}(n) = \begin{cases} k - i, & \text{if } n \in V_k \text{ and } n \text{ is the top } i \text{ node,} \\ 0, & \text{otherwise.} \end{cases} \tag{6}$$

Edge prizes are assigned similarly. The objective is to identify a subgraph, $S^* = (V^*, E^*)$, that optimizes the total prize of nodes and edges, minus the costs associated with the size of the subgraph:

$$S^* = \underset{\substack{S \subseteq G, \\ S \text{ is connected}}}{\operatorname{argmax}} \sum_{n \in V_S} \text{prize}(n) + \sum_{e \in E_S} \text{prize}(e) - \text{cost}(S), \tag{7}$$

where

$$\text{cost}(S) = |E_S| \times C_e, \tag{8}$$

and $C_e$ denotes a predefined cost per edge, which is adjustable to control the subgraph size.

The original PCST algorithm is designed for node prizes only. However, given the significance of edge semantics in certain scenarios, we adapt the algorithm to accommodate edge prizes as follows: Consider an edge e with a cost $C_e$ and a prize $P_e$. If $C_e > P_e$, it can be treated as a reduced edge cost of $C_e - P_e$. However, if $P_e > C_e$, negative edge costs are not allowed in the original algorithm. Our solution involves replacing edge $e$ with a 'virtual node' $v_e$, connected to both endpoints of $e$. This virtual node is assigned a prize of $P_e - C_e$, and the cost of the two new edges leading to the virtual node is set to zero. This modification effectively mirrors the original problem, as including edge $e$ in the original graph is analogous to including the virtual node in the modified graph. Finally, we optimize the PCST problem using a near-linear time approach [9].

## 5.4 Answer Generation

**Graph Encoder.** Let $S^* = (V^*, E^*)$ represent the retrieved subgraph. We use a graph encoder to model the structure of this graph, specifically using a standard Graph Attention Network (GAT) [43]. Our approach for encoding the retrieved subgraph is defined as follows:

$$h_g = \text{POOL}(\text{GNN}_{\phi_1}(S^*)) \in \mathbb{R}^{d_g}, \tag{9}$$

Here, POOL denotes the mean pooling operation, and $d_g$ is the dimension of the graph encoder.

**Projection Layer.** We incorporate a multilayer perceptron (MLP) to align the graph token with the vector space of the LLM:

$$\hat{h}_g = \text{MLP}_{\phi_2}(h_g) \in \mathbb{R}^{d_l}, \tag{10}$$

where $d_l$ is the dimension of the LLM's hidden embedding.

**Text Embedder.** To leverage the text-reasoning capabilities of LLMs, we transform the retrieved subgraph $S^*$ into a textual format. This transformation involves flattening the textual attributes of the

nodes and edges, as illustrated in the green box in Figure 2. We refer to this operation as textualize($\cdot$). Subsequently, we combine the textualized graph with the query to generate a response. Let $x_q$ denote the query; we concatenate it with the textualized graph textualize($S^*$). We then map the result to an embedding $h_t$ using a text embedder, which is the first layer of a pretrained and frozen LLM:

$$h_t = \text{TextEmbedder}([\text{textualize}(S^*); x_q]) \in \mathbb{R}^{L \times d_l}, \tag{11}$$

where $[;]$ represents the concatenation operation, and $L$ is the number of tokens.

**LLM Generation with Graph Prompt Tuning.** The final stage involves generating the answer $Y$ given the graph token $\hat{h}_g$, acting as a soft prompt, and the text embedder output $h_t$. These inputs are fed through the self-attention layers of a pretrained frozen LLM, with parameter $\theta$. The generation process is represented as follows:

$$p_{\theta,\phi_1,\phi_2}(Y|S^*, x_q) = \prod_{i=1}^{r} p_{\theta,\phi_1,\phi_2}(y_i|y_{<i}, [\hat{h}_g; h_t]), \tag{12}$$

where $[\hat{h}_g; h_t]$ concatenates the graph token $\hat{h}_g$ and the text embedder output $h_t$. While $\theta$ is frozen, the graph token $\hat{h}_g$ receives gradients, enabling the optimization of the parameters of the graph encoder $\phi_1$ and the projection layer $\phi_2$ through standard backpropagation.

## 6 Experiments

### 6.1 Experiment Setup

In the indexing step, we use SentenceBert [34] as the LM to encode all node and edge attributes. In the generation step, we use the open-source Llama2-7b [42] as the LLM and Graph Transformer [37] as the graph encoder. Additional details are provided in Appendix B.1.

### 6.2 Main Results

In our experiments, we consider three model configurations: *1) Inference-only*: Using a frozen LLM for direct question answering; *2) Frozen LLM w/ prompt tuning (PT)*: Keeping the parameters of the LLM frozen and adapting only the prompt; *3) Tuned LLM*: Fine-tuning the LLM with LoRA [10]. We provide more details in Appendix B.2.

Table 3: Performance comparison across `ExplaGraphs`, `SceneGraphs`, and `WebQSP` datasets for different configurations, including Inference-only, Frozen LLM with prompt tuning (PT), and Tuned LLM settings. Mean scores and standard deviations (mean ± std) are presented. The first best result for each task is highlighted in **bold** and the second best result is highlighted with an underline.

| Setting | Method | ExplaGraphs | SceneGraphs | WebQSP |
|---|---|---|---|---|
| Inference-only | Zero-shot | 0.5650 | 0.3974 | 41.06 |
| | Zero-CoT [18] | 0.5704 | 0.5260 | 51.30 |
| | CoT-BAG [44] | 0.5794 | 0.5680 | 39.60 |
| | KAPING [1] | 0.6227 | 0.4375 | 52.64 |
| Frozen LLM w/ PT | Prompt tuning | 0.5763 ± 0.0243 | 0.6341 ± 0.0024 | 48.34 ± 0.64 |
| | GraphToken [31] | 0.8508 ± 0.0551 | 0.4903 ± 0.0105 | 57.05 ± 0.74 |
| | *G-Retriever* | 0.8516 ± 0.0092 | 0.8131 ± 0.0162 | 70.49 ± 1.21 |
| | $\Delta_{\text{Prompt tuning}}$ | ↑ 47.77% | ↑ 28.23% | ↑ 45.81% |
| Tuned LLM | LoRA | 0.8538 ± 0.0353 | 0.7862 ± 0.0031 | 66.03 ± 0.47 |
| | *G-Retriever* w/ LoRA | **0.8705 ± 0.0329** | **0.8683 ± 0.0072** | **73.79 ± 0.70** |
| | $\Delta_{\text{LoRA}}$ | ↑ 1.95% | ↑ 11.74% | ↑ 10.44% |

Table 3 demonstrates the effectiveness of our method across three datasets in various configurations. In the inference-only setting, *G-Retriever* surpasses all baselines. Notably, LLM can perform even better when no graph knowledge is provided (*i.e.,* question only), which might be attributed to the complexity and potential noise in the knowledge. For frozen LLM with prompt tuning, *G-Retriever* outperforms traditional prompt tuning and GraphToken [31], a graph prompt tuning-based method, with average performance increases of 40.6% and 30.8% respectively. Furthermore, when tuned with LoRA, *G-Retriever* achieves the best performance.

Table 4: Retrieval on graphs significantly improves efficiency.

| Dataset | Before Retrieval (Avg.) | | | After Retrieval (Avg.) | | |
|---|---|---|---|---|---|---|
| | # Tokens | # Nodes | Min/Epoch | # Tokens | # Nodes | Min/Epoch |
| SceneGraphs | 1,396 | 19 | 123.1 | 235 (↓83%) | 5 (↓74%) | 86.8 (↓29%) |
| WebQSP | 100,627 | 1,371 | 18.7 | 610 (↓99%) | 18 (↓99%) | 6.2(↓67%) |

## 6.3 Efficiency Evaluation

The efficiency of our approach is highlighted by the data in Table 4. Implementing our graph-based retrieval significantly decreases the number of tokens required to describe the graphs in text, reduces the number of nodes in graphs, and speeds up the training process. Specifically, for the SceneGraphs dataset, tokens decreased by 83%, nodes by 74%, and training time by 29%. For the WebQSP dataset, tokens decreased by 99%, nodes by 99%, and training time by 67%. These substantial reductions demonstrate the method's efficiency and potential in managing large-scale graph data.

## 6.4 Mitigation of Hallucination

To evaluate hallucination, we instructed the models to answer graph-related questions, specifically by identifying supporting nodes or edges from the graph. We assessed the model's faithfulness using three metrics: the fraction of valid nodes (denoted as Valid Nodes), the fraction of valid edges (denoted as Valid Edges), and the fraction of times the entire set of cited nodes and edges was valid (denoted as Fully Valid Graphs). We manually reviewed 100 responses from both our method and the baseline (*i.e.,*, LLM with graph prompt tuning). Table 5 shows that *G-Retriever* significantly reduces hallucinations by 54% compared to the baseline, as our graph retrieval ensures that the data is sourced directly from the actual graph, leading to fewer hallucinations. See Appendix G for details.

## 6.5 Ablation Study

In this ablation study, we assess the individual impact of key components within our pipeline. As shown in Table 6, there are performance drops when any of these components are removed, with the graph encoder and textualized graph showing declines of 22.51% and 19.19%, respectively. This demonstrates their complementary effects in representing the graph in both textual and embedded formats. Additionally, the retrieval on graphs is also important to the overall performance. Further details are available in Appendix B.3. We also present additional studies on our framework: it is robust to the choice of graph encoders (see Appendix B.4) and benefits from the increased scale of LLMs (see Appendix B.5).

Table 5: Hallucination reduction on the SceneGraphs dataset, measured by fractions of valid nodes, valid edges, and fully valid graphs (where all nodes and edges are correct).

| | Baseline | *G-Retriever* |
|---|---|---|
| Valid Nodes | 31% | 77% |
| Valid Edges | 12% | 76% |
| Fully Valid Graphs | 8% | 62% |

Table 6: Ablation study on the WebQSP dataset showing performance drops (Hit@1) when each component is removed.

| Method | Hit@1 |
|---|---|
| *G-Retriever* | 70.49 |
| w/o Graph Encoder | 54.62 (↓22.51%) |
| w/o Projection Layer | 69.70 (↓1.11%) |
| w/o Textualized Graph | 56.96 (↓19.19%) |
| w/o Retrieval | 63.84 (↓9.43%) |

Additionally, we include a detailed comparison with existing retrieval methods (see Appendix D), a discussion on the complexity (see Appendix E), and demonstrations on how to use *G-Retriever* to 'chat with your graph' (see Appendix H).

# 7 Conclusion

In this work, we introduce a new GraphQA benchmark for real-world graph question answering and present *G-Retriever*, an architecture adept at complex and creative queries. Experimental results show that *G-Retriever* surpasses baselines in textual graph tasks across multiple domains, scales effectively with larger graph sizes, and demonstrates resistance to hallucination.

**Limitations and Future Work:** Currently, *G-Retriever* employs a static retrieval component. Future developments could investigate more sophisticated RAG where the retrieval is trainable.

# Acknowledgment

BH is supported by the Ministry of Education, Singapore, under the Academic Research Fund Tier 1 (FY2023) (Grant A-8001996-00-00). XB is supported by NUS Grant ID R-252-000-B97-133. The authors would like to express their gratitude to the reviewers for their feedback, which has improved the clarity and contribution of the paper.

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

# A Impact Statements

As LLMs are applied to increasingly diverse tasks, their ability to process complex structured data will be increasingly vital. Our work aims to enhance LLMs' ability to interact with graph-structured data, while resisting hallucination, thus improving model reliability. We also enhance explainability, both by returning the retrieved subgraph, and through the use of conversational interfaces for 'chatting with a graph', which allows for better human-AI interaction and for models to behave in a way that is more well-aligned with human expectations.

# B Experiment

## B.1 Implementation Settings

Experiments are conducted using 2 NVIDIA A100-80G GPUs. Each experiment is replicated four times, utilizing different seeds for each run to ensure robustness and reproducibility.

**Graph Encoder.** We use Graph Transformer [37] as the GNN backbone. Our configuration employs 4 layers, each with 4 attention heads, and a hidden dimension size of 1024.

**LLM.** We use the open-sourced Llama2-7b [42] as the LLM backbone. In fine-tuning the LLM with LoRA [10], the lora_r parameter (dimension for LoRA update matrices) is set to 8, and lora_alpha (scaling factor) is set to 16. The dropout rate is set to 0.05. In prompt tuning, the LLM is configured with 10 virtual tokens. The number of max text length is 512, the number of max new tokens, *i.e.,* the maximum numbers of tokens to generate, is 32.

**PCST**. For retrieval over graphs via PCST, for the `SceneGraphs` dataset, we select the top $k$ nodes and edges, setting $k$ to 3. Here, the cost of edges, denoted as $C_e$, is set to 1. Regarding the `WebQSP` dataset, we set $k = 3$ for nodes and $k = 5$ for edges, with the edge cost, $C_e$, adjusted to 0.5. For the `ExplaGraphs` dataset, which is characterized by a small graph size averaging 5.17 nodes and 4.25 edges (as detailed in Table 2), the entire graph can fit in the LLM's context window size. Consequently, we aim to retrieve the whole graph by setting $k$ to 0, effectively returning the original graph unaltered.

**Optimization.** We use the AdamW [27] optimizer. We set the initial learning rate at 1e-5, with a weight decay of 0.05. The learning rate decays with a half-cycle cosine decay after the warm-up period. The batch size is 4, and the number of epochs is 10. To prevent overfitting and ensure training efficiency, an early stopping mechanism is implemented with a patience setting of 2 epochs.

## B.2 Details of Model Configurations

In our experiments, we consider three model configurations:

*1) Inference-only*: Using a frozen LLM for direct question answering with textual graph and question, see Figure 4.

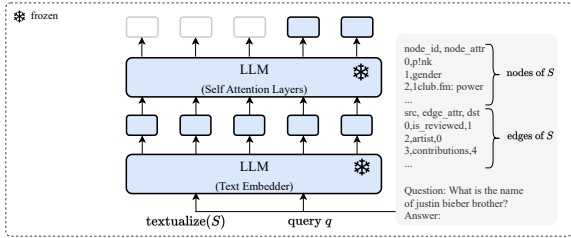

Figure 4: Model configuration *1) Inference-only*.

- Zero-shot. In this approach, the model is given a textual graph description and a task description, and is immediately asked to produce the desired output. No additional examples or demonstrations are provided.

- Zero-CoT. Zero-shot Chain-of-thought (Zero-CoT) prompting [18] is a follow-up to CoT prompting [45], which introduces an incredibly simple zero shot prompt by appending the words "Let's think step by step." to the end of a question.

- CoT-BAG. Build-a-Graph Prompting (BAG) [44] is a prompting technique that adds "Let's construct a graph with the nodes and edges first." after the textual description of the graph is explicitly given.

- KAPING. KAPING [1] is a zero-shot knowledge-augmented prompting method for knowledge graph question answering. It first retrieves triples related to the question from the graph, then prepends them to the input question in the form of a prompt, which is then forwarded to LLMs to generate the answer.

*2) Frozen LLM w/ prompt tuning (PT)*: Keeping the parameters of the LLM frozen and adapting only the prompt. This includes soft prompt tuning (see Figure 5a), GraphToken [31], which is a graph prompt tuning method, and our *G-Retriever* method (see Figure5b).

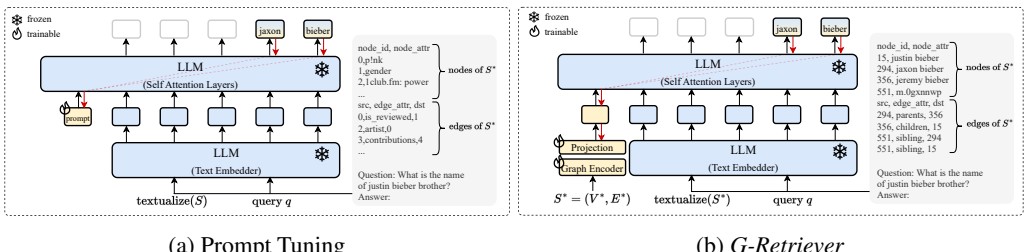

(a) Prompt Tuning  (b) *G-Retriever*

Figure 5: Model configuration *2) Frozen LLM w/ prompt tuning*.

*3) Tuned LLM*: Fine-tuning the LLM with LoRA. This includes standard fine-tuning of an LLM for downstream tasks using LoRA (see Figure 6a) and G-Retriever with LoRA (see Figure 6b).

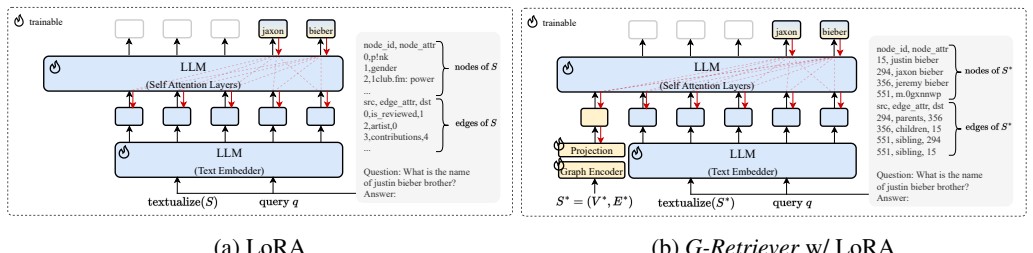

(a) LoRA  (b) *G-Retriever* w/ LoRA

Figure 6: Model configuration *3) Tuned LLM*.

## B.3 Details of Ablation Study

This section illustrates the modifications made to the original architecture in the ablation study, as presented in Figure 7.

**Without Graph Encoder (w/o GraphEncoder):** In this setting, we replaced the graph encoder with trainable soft tokens, setting the number of these virtual tokens to 10.

**Without Projection Layer (w/o Projection Layer):** Here, we removed the projection layer following the graph encoder. We configured the output dimension of the graph encoder to be 4,096, matching the hidden dimension of Llama2-7b. This allows the output graph token (the yellow token in Figure 7b) to be concatenated directly with the LLM tokens (blue tokens).

**Without Textualized Graph (w/o Textualized Graph):** In this configuration, we modified the textual input to the LLM. Rather than using a combination of the question and the textualized graph, we solely used the question.

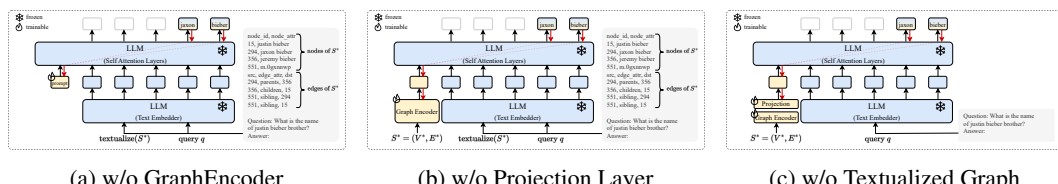

| (a) w/o GraphEncoder | (b) w/o Projection Layer | (c) w/o Textualized Graph |

Figure 7: Ablation study configurations.

## B.4 The Choice of Graph Encoder

In addition to the Graph Transformer [37], we explore other GNNs as the graph encoder, such as GCN [17] and the GAT [43]. The comparative results of these models on the `WebQSP` and `ExplaGraphs` datasets are presented in Table 7.

Table 7: Performance of different graph encoders on the `WebQSP` and `ExplaGraphs` datasets.

| Graph Encoder | WebQSP | ExplaGraphs |
|---|---|---|
| GCN [17] | 70.70 | 0.8394 |
| GAT [43] | 70.27 | 0.8430 |
| Graph Transformer [37] | 70.49 | 0.8516 |

The results demonstrate that our proposed method exhibits consistent robustness across different graph encoders. Notably, all three encoders – GCN, GAT, and GraphTransformer – demonstrate competitive and closely aligned performance on the `WebQSP` dataset, with Hit@1 scores of 70.70, 70.27, and 70.49, respectively. However, the performance differentiation becomes more pronounced on the `ExplaGraphs` dataset, where GraphTransformer exhibits a superior Hit@1 score of 0.8516, followed by GAT and GCN with scores of 0.8430 and 0.8394, respectively. This variation in performance across the datasets highlights the importance of encoder selection based on the specific characteristics and requirements of the dataset.

## B.5 The Choice of LLM

As for the choice of LLM, we considered both Llama2-7b and Llama2-13b. Our experiments demonstrate that stronger LLMs enhance the effectiveness of our method, as shown in Table 8, indicating that it benefits from the increased scale of the LLMs.

Table 8: Performance of different LLMs on the `WebQSP` dataset.

| LLM | Llama2-7b | Llama2-13b |
|---|---|---|
| Hit@1 | 70.49 | 75.58 |

## C GraphQA Benchmark

In this section, we detail how our GraphQA benchmark differs from the original datasets, including the specific processing steps we employed. For concrete examples that illustrate the differences between the raw text in the original dataset and in our GraphQA benchmark, please refer to Table 9.

`ExplaGraphs`. The original dataset[1] [35] represents relationships using triplets. We have standardized this format by converting the triplets into a graph representation. Specifically, each head and tail in a triplet is transformed into a node, and the relation is transformed into an edge. Since the test dataset labels are not available, we have utilized only the training and validation (val) datasets from the original collection. We further divided these into training, val, and test subsets, using a 6:2:2 ratio.

---

[1]https://explagraphs.github.io/

Table 9: Comparison of text formats in original datasets and our GraphQA benchmark.

| Dataset | Original dataset | GraphQA Benmark |
|---|---|---|
| ExplaGraphs | (entrapment; capable of; being abused) (being abused; created by; police) (police; capable of; harm) (harm; used for; people) (people; part of; citizens) | node_id,node_attr\n 0,entrapment\n 1,being abused\n 2,police\n 3,harm\n 4,people\n 5,citizens\n
src,edge_attr,dst\n 0,capable of,1\n 1,created by,2\n 2,capable of,3\n 3,used for,4\n 4,part of,5\n |
| SceneGraphs | "width": 500, "objects": {"681267": {"name": "banana", "h": 34, "relations": [{"object": "681262", "name": "to the left of"}], "w": 64, "attributes": ["small", "yellow"], "y": 55, "x": 248, "681265": {"name": "spots", "h": 16, "relations": [], "w": 26, "attributes": [], "y": 92, "x": 245, "681264": {"name": "bananas", "h": 50, "relations": [{"object": "681259", "name": "to the left of"}], "w": 49, "attributes": ["small", "yellow"], "y": 32, "x": 268, "681263": {"name": "picnic", "h": 374, "relations": [], "w": 499, "attributes": ["delicious"], "y": 0, "x": 0, "681262": {"name": "straw", "h": 95, "relations": [{"object": "681268", "name": "to the right of"}, {"object": "681267", "name": "to the right of"}, {"object": "681253", "name": "to the right of"}], "w": 15, "attributes": ["white", "plastic"], "y": 55, "x": 402, "681261": {"name": "meat", "h": 27, "relations": [{"object": "681255", "name": "on"}, {"object": "681255", "name": "inside"}], "w": 24, "attributes": ["small", "brown", "delicious"], "y": 123, "x": 68, "681260": {"name": "rice", "h": 57, "relations": [{"object": "681255", "name": "on", "object": "681258", "name": "to the left of"}], "w": 93, "attributes": ["piled", "white"], "y": 162, "x": 57, "681269": {"name": "onions", "h": 16, "relations": [], "w": 24, "attributes": ["green"], "y": 147, "x": 90, "681268": {"name": "tablecloth", "h": 374, "relations": [{"object": "681262", "name": "to the left of"}], "w": 396, "attributes": ["white"], "y": 0, "x": 0, "681258": {"name": "bowl", "h": 99, "relations": [{"object": "681255", "name": "next to", "object": "681257", "name": "of", "object": "681255", "name": "near", "object": "681256", "name": "to the right of", "object": "681260", "name": "to the right of", "object": "681255", "name": "to the right of"}], "w": 115, "attributes": ["full"], "y": 184, "x": 178, "681259": {"name": "plantains", "h": 70, "relations": [{"object": "681264", "name": "to the right of"}], "w": 45, "attributes": ["red"], "y": 0, "x": 346, "681256": {"name": "spoon", "h": 65, "relations": [{"object": "681255", "name": "on", "object": "681257", "name": "to the left of", "object": "681255", "name": "in", "object": "681258", "name": "to the left of"}], "w": 140, "attributes": ["large", "metal", "silver"], "y": 196, "x": 0, "681257": {"name": "dish", "h": 81, "relations": [{"object": "681258", "name": "inside", "object": "681258", "name": "to the right of", "object": "681258", "name": "in", "object": "681255", "name": "to the right of"}], "w": 108, "attributes": ["cream colored"], "y": 199, "x": 187, "681254": {"name": "meat", "h": 111, "relations": [], "w": 130, "attributes": [], "y": 121, "x": 58, "681255": {"name": "plate", "h": 138, "relations": [{"object": "681257", "name": "to the left of", "object": "681254", "name": "of", "object": "681254", "name": "with", "object": "681258", "name": "near", "object": "681258", "name": "to the left of"}], "w": 176, "attributes": ["white", "full"], "y": 111, "x": 30, "681253": {"name": "banana", "h": 30, "relations": [{"object": "681262", "name": "to the left of"}], "w": 73, "attributes": ["small", "yellow"], "y": 87, "x": 237, "height": 375 | node_id,node_attr
0,"name: banana; attribute: small, yellow; (x,y,w,h): (248, 55, 64, 34)"
1,"name: spots; (x,y,w,h): (245, 92, 26, 16)"
2,"name: bananas; attribute: small, yellow; (x,y,w,h): (268, 32, 49, 50)"
3,"name: picnic; attribute: delicious; (x,y,w,h): (0, 0, 499, 374)"
4,"name: straw; attribute: white, plastic; (x,y,w,h): (402, 55, 15, 95)"
5,"name: meat; attribute: small, brown, delicious; (x,y,w,h): (68, 123, 24, 27)"
6,"name: rice; attribute: piled, white; (x,y,w,h): (57, 162, 93, 57)"
7,"name: onions; attribute: green; (x,y,w,h): (90, 147, 24, 16)"
8,"name: tablecloth; attribute: white; (x,y,w,h): (0, 0, 396, 374)"
9,"name: bowl; attribute: full; (x,y,w,h): (178, 184, 115, 99)"
10,"name: plantains; attribute: red; (x,y,w,h): (346, 0, 45, 70)"
11,"name: spoon; attribute: large, metal, silver; (x,y,w,h): (0, 196, 140, 65)"
12,"name: dish; attribute: cream colored; (x,y,w,h): (187, 199, 108, 81)"
13,"name: meat; (x,y,w,h): (58, 121, 130, 111)"
14,"name: plate; attribute: white, full; (x,y,w,h): (30, 111, 176, 138)"
15,"name: banana; attribute: small, yellow; (x,y,w,h): (237, 87, 73, 30)"
src,edge_attr,dst
0,to the left of,4\n 2,to the left of,10\n 4,to the right of,8\n 4,to the right of,0\n 4,to the right of,15\n 5,on,14\n 5,inside,14\n 6,on,14\n 6,to the left of,9\n 8,to the left of,4\n 9,next to,14\n 9,of,12\n 9,near,14\n 9,to the right of,11\n 9,to the right of,6\n 9,to the right of,14\n 10,to the right of,2\n 11,on,14\n 11,to the left of,12\n 11,in,14\n 11,to the left of,9\n 12,inside,9\n 12,to the right of,11\n 12,in,9\n 12,to the right of,14\n 14,to the left of,12\n 14,of,13\n 14,with,13\n 14,near,9\n 14,to the left of,9\n 15,to the left of,4\n |
| WebQSP | [['FedEx Cup', 'sports.sports_award_type.winners', 'm.0n1v8cy'], ['Brandt Snedeker', 'sports.sports_award_winner.awards', 'm.0n1v8cy'], ['FedEx Cup', 'common.topic.article', 'm.08q5wy'], ['FedEx Cup', 'common.topic.notable_for', 'g.12559n8g_'], ['Sports League Award Type', 'freebase.type_profile.published', 'Published'], ['FedEx Cup', 'common.topic.notable_types', 'Sports League Award Type'], ['m.0n1v8cy', 'sports.sports_award.award_winner', 'Brandt Snedeker'], ['Sports League Award Type', 'type.type.expected_by', 'Award'], ['Sports League Award Type', 'common.topic.article', 'm.06zxtxj'], ['2012 PGA Tour', 'sports.sports_league_season.awards', 'm.0n1v8cy'], ['Sports League Award Type', 'freebase.type_hints.included_types', 'Topic'], ['Sports League Award Type', 'type.type.domain', 'Sports'], ['m.0n1v8cy', 'sports.sports_award.award', 'FedEx Cup'], ['Sports League Award Type', 'freebase.type_profile.strict_included_types', 'Topic'], ['Sports League Award Type', 'freebase.type_profile.kind', 'Classification'], ['m.0n1v8cy', 'sports.sports_award.season', '2012 PGA Tour'], ['Sports League Award Type', 'type.type.properties', 'Winners']] | node_id,node_attr\n 0,fedex cup\n 1,m.0n1v8cy\n 2,brandt snedeker\n 3,m.08q5wy\n 4,g.12559n8g_\n 5,sports league award type\n 6,published\n 7,award\n 8,m.06zxtxj\n 9,2012 pga tour\n 10,topic\n 11,sports\n 12,classification\n 13,winners\n
src,edge_attr,dst
0,sports.sports_award_type.winners,1
2,sports.sports_award_winner.awards,1
0,common.topic.article,3
0,common.topic.notable_for,4
5,freebase.type_profile.published,6
0,common.topic.notable_types,5
1,sports.sports_award.award_winner,2
5,type.type.expected_by,7
5,common.topic.article,8
9,sports.sports_league_season.awards,1
5,freebase.type_hints.included_types,10
5,type.type.domain,11
1,sports.sports_award.award,0
5,freebase.type_profile.strict_included_types,10
5,freebase.type_profile.kind,12
1,sports.sports_award.season,9
5,type.type.properties,13 |

SceneGraphs. The original GQA dataset is designed for real-world visual reasoning and compositional question answering, aiming to address key shortcomings of previous VQA datasets [13]. It comprises 108k images, each associated with a Scene Graph. In our study, we focus differently on graph question answering; hence, we did not utilize the image counterparts, leveraging only the scene graphs from the original dataset. Additionally, the original dataset describes images using JSON files. We simplified the object IDs to suit our research needs. We randomly sampled 100k samples from the original dataset and divided them into training, validation, and test subsets, following a 6:2:2 ratio.

WebQSP. We follow the preprocessing steps from RoG[2] [28]. The original dataset uses a list of triplets format, which we have transformed into our unified graph format. Furthermore, to avoid

---

[2]https://huggingface.co/datasets/rmanluo/RoG-webqsp

discrimination between capital and lowercase words, we have converted all words to lowercase. We used the same dataset split as in the original dataset.

**Contribution of the GraphQA Benchmark.** We acknowledge that the GraphQA benchmark involves converting three existing graph datasets into a uniform format. However, we believe this standardization provides significant value to the research community in several ways:

- **Task Introduction:** Unlike existing graph question-answering benchmarks that focus on small or synthetic graphs, our benchmark includes real-world applications and frames them as graph question-answering tasks.

- **Standardization:** A key and significant effort of this benchmark is the standardization and processing of diverse datasets into a uniform format suitable for GraphQA tasks. These datasets, previously used in different contexts, are redesigned to focus specifically on GraphQA, ensuring consistent and comparable evaluations across models.

- **Accessibility:** We have open-sourced the GraphQA benchmark, providing a unified format that simplifies model application across multiple datasets. This reduces the complexity of handling various data structures and preprocessing pipelines, lowering barriers for new researchers and encouraging broader participation. We have already seen several novel works using our GraphQA benchmark, and we expect rapid adoption within the LLM and GNN communities.

- **Baseline Comparisons:** The benchmark offers baseline performance metrics, helping researchers identify the strengths and weaknesses of new approaches compared to established baselines.

# D   Graph Retrieval-Augmented Generation (GraphRAG)

## D.1   Elaboration on PCST-Based Retrieval

**Modeling motivation.** We formulate subgraph retrieval as a Prize-Collecting Steiner Tree (PCST) optimization problem. This is motivated by the need to find a connected subgraph containing most relevant nodes and edges, a goal that aligns well with the objectives of PCST: maximizing node values while minimizing edge costs. Though not universally acknowledged as optimal, we have empirically validated its effectiveness.

**Effectiveness compared to other retrieval baselines.** To validate the effectiveness of our PCST-based retrieval approach, we compared it against several baselines: (1) top-k triples retrieval, *i.e.,* KAPING [1], which retrieves the top-k triples related to the query and incorporates them into the prompt for the LLM; (2) top-k nodes plus neighbors, which retrieves the top-k nodes and their one-hop neighbors, capturing local context; (3) shortest path retrieval, which retrieves the top-k nodes and computes the shortest paths between them.

For all methods, we set $k = 5$ and used llama2-7b-chat as the LLM. The results, presented in Table 10, show that our PCST-based retrieval method achieves the highest accuracy (Hit@1) of 66.17% on the WebQSP dataset, outperforming all baseline methods.

Table 10: Comparison of retrieval methods on the `WebQSP` dataset.

| Method | Hit@1 |
|---|---|
| PCST retrieval | 66.17 |
| top-k triples retrieval (KAPING) | 52.64 |
| top-k nodes plus its neighbors | 49.82 |
| shortest path retrieval | 55.20 |

## D.2   Advantages of Subgraph-Based Retrieval

**Context-Relevant.** Selecting nodes and edges in isolation may overlook neighborhood information. In contrast, PCST-based retrieval is guaranteed to return a connected subgraph, capturing the graph

context during the retrieval process. This approach retrieves not only high-relevance nodes or edges but also "bridge" elements that connect these with contextually significant nodes or edges, which are crucial for generating a comprehensive response.

**Size Management.** Compared to the shortest path method, PCST retrieval provides greater control over the size of the retrieved subgraph. By adjusting the prizes and costs on nodes and edges, users can fine-tune the subgraph's extent. In contrast, the shortest path approach lacks the ability to control the distance between the top-k nodes, which can lead to disconnected subgraphs or the inclusion of unnecessarily long paths.

### D.3 The Impact of K for Retrieval

We identify the most relevant nodes and edges and use a k-nearest neighbors retrieval approach (see Equation 6). Small k values may omit crucial knowledge or information relevant to the query, while large k values could introduce excessive information, distracting the model from the essential details. To evaluate the impact of the number of k, we have conducted additional experiments by varying the choice of k to 3, 5, 10, and 20.

Table 11: The impact of k on the `webqsp` dataset.

| k | 3 | 5 | 10 | 20 |
|---|---|---|---|---|
| Hit@1 | 0.6977 | 0.7063 | 0.7248 | 0.7039 |

As shown in Table 11, the Hit@1 metric initially rises for small k values, peaks at a certain point, and then declines for large k values. Determining the optimal k value can be achieved through techniques like cross-validation using a validation set.

### D.4 The Choice of Similarity Function

The choice of similarity function is also important. In this work, we use cosine similarity, a widely adopted metric for measuring vector similarity in models that process vision and language. For instance, CLIP also employs cosine similarity to assess the similarity between text and image features. Although it might not be the optimal choice, we believe that cosine similarity is a general, representative, and valid choice for facilitating fast retrieval tasks.

## E Discussion on the Complexity

### E.1 The integration of GNNs, LLMs and GraphRAG

*G-Retriever* is framework integrate the strengths of GNNs, LLMs and GraphRAG. The LLM+X framework, which involves enriching LLMs with multi-modal capabilities by integrating an LLM with an encoder from another modality, is a widely adopted approach. Notable examples include Llava, MiniGPT-4, and Flamingo, among others. They are not complex in terms of understanding or implementation. Regarding the integration of GraphRAG, it does not require training and can be implemented during the preprocessing stage or on the fly. This approach does not significantly increase time complexity or computational complexity. On the contrary, it can substantially reduce the size of the graph (*e.g.,* eliminating 99% of nodes in the `WebQSP` dataset), which in turn speeds up the overall running time (*e.g.,* reducing it from 18.7 min/epoch to 6.2 min/epoch on the `WebQSP` dataset).

### E.2 Computational Resources

Utilizing two A100 GPUs, each with 80GB of memory, we conducted tests on Llama2-7b and `WebQSP` datasets. Our experiments had a training batch size of 16 and an evaluation batch size of 32, yielding the following results.

These results highlight efficiency improvements via graph RAG, which significantly reduces graph size (e.g., eliminating 99% of nodes in the `WebQSP` dataset) and speeds up running time.

Table 12: Performance and Efficiency of Various Methods on the `WebQSP` dataset.

| Settting | Method | Hit@1 | Time |
|---|---|---|---|
| Inference-only | Question only | 61.16 | 31 min |
|  | Textual graph and question | 41.06 | 40 min |
| Frozen LLM w/ PT | Prompt Tuning | 48.34 | 18.7 min/epoch |
|  | G-Retriever | 70.49 | 6.2 min/epoch |
| Tuned LLM | LoRA | 66.03 | 19 min/epoch |
|  | *G-Retriever* w/ LoRA | 73.79 | 6.9 min/epoch |

# F Discussion on Explainability

We believe G-Retriever enhances explainability in the following ways:

**Retrieved subgraph.** By returning the most relevant subgraph in response to a query, users can see which parts of the graph are considered important for the answer. This helps users understand the basis of the model's responses. For example, if users want to understand why certain information is present or absent in the LLM's response, they can inspect the subgraph to see whether such information is present or absent in the retrieved subgraph.

**Conversational Interface.** G-Retriever allows users to ask follow-up questions and receive detailed natural language explanations. For example, if a user questions the LLM's response, they can ask, "Why do you think [xxx]? Please explain your answer." This interactive capability enables users to explore the model's reasoning process and gain deeper insights into how it interprets graph data.

# G Hallucination in Graph LLMs

In this section, we present quantitative results regarding hallucinations in the `SceneGraphs` dataset.

**Baseline.** For our baseline, we adapted MiniGPT-4 [57] to graph contexts. This approach involves a frozen LLM interacting with a trainable GNN that encodes graph data as a soft prompt, denoted as LLM+Graph Prompt Tuning. We focus on graph prompt tuning as the baseline, instead of converting the graph into text, since the textual representation of the graph is large and consistently exceeds the input token limits of LLMs.

**Experiment Design.** We instructed the LLM to answer graph-related questions and to list nodes or edges in the explanation graph that support its answers. Since standard answers for these questions do not exist, allowing the LLM to respond flexibly, it becomes challenging to evaluate its responses. To address this, we manually examined 100 responses generated by our method and the LLM with graph prompt tuning, verifying whether the nodes and edges referenced in the LLM's output actually exist in the graph.

**Evaluation Metrics.** We assessed the model's faithfulness using three metrics: the fraction of valid nodes (denoted as Valid Nodes), the fraction of valid edges (denoted as Valid Edges), and the fraction of times the entire set of nodes and edges cited was valid (denoted as Fully Valid Graphs).

**Results.** The results, as depicted in Table 5, illustrate the comparative effectiveness of the *G-Retriever* over the baseline LLM+Graph Prompt Tuning method in reducing hallucinations. The LLM+Graph Prompt Tuning approach demonstrated a significantly lower accuracy in referencing graph elements, with only 31% of nodes and 12% of edges being valid, and the entire set of nodes and edges being valid only 8% of the time. In contrast, *G-Retriever* showed substantial improvements: 77% validity in nodes, 76% in edges, and 62% in the overall validity of referenced node-edge sets. These results underscore the significant reduction in hallucinations with *G-Retriever*, particularly in the challenging task of accurately citing both nodes and edges in graph-based contexts.

# H Demonstrations

We demonstrate the interaction capabilities of G-Retriever with creative questions on different datasets: `ExplaGraphs`, `SceneGraphs`, and `WebQSP`. These examples are showcased in Tables 13, 14, and 15, respectively. Additionally, the examples are visualized in Figure 1.

Table 13: Example of creative chat on `ExplaGraphs`.

| Graph | node_id,node_attr
0,human
1,cosmetic surgery
2,women
3,change appearance
4,addiction
5,searching for perfection
6,stop
7,positive

src,edge_attr,dst
0,receives action,1
2,part of,0
1,used for,3
3,capable of,4
4,created by,5
4,not capable of,6
4,is not a,7 |
|---|---|
| User | Write a 500-word argument essay based on the information provided in the explanation graph. |
| Model | The explanation graph provides a detailed analysis of the reasons why women undergo cosmetic surgery. According to the graph, the main reasons include the desire to change one's appearance, the influence of societal beauty standards, and the addiction to searching for perfection. These reasons are further supported by the various nodes and edges in the graph, which provide additional insights into the motivations and consequences of cosmetic surgery.
Firstly, the node labeled "human" highlights the fundamental human desire to feel attractive and confident in one's appearance. This desire is deeply ingrained in our culture, where beauty is often associated with happiness, success, and social acceptance. As a result, many women feel pressure to conform to societal beauty standards, which can lead to feelings of insecurity and low self-esteem. In response to these feelings, some women may turn to cosmetic surgery as a means of improving their appearance and boosting their self-confidence.
Secondly, the node labeled "women" highlights the gendered nature of cosmetic surgery. While men also undergo cosmetic surgery, the majority of cosmetic surgery patients are women. This is likely due to the fact that women are more likely to be exposed to societal beauty standards and are more likely to internalize these standards. As a result, women may be more likely to feel insecure about their appearance and more likely to seek cosmetic surgery as a means of improving their looks.
Thirdly, the node labeled "addiction" highlights the potential negative consequences of cosmetic surgery. While some women may initially feel improved confidence and self-esteem after undergoing cosmetic surgery, the graph suggests that this initial boost may not last. In fact, some women may become addicted to the constant pursuit of perfection, leading to a cycle of surgery and dissatisfaction. This can have serious psychological and emotional consequences, including depression, anxiety, and a loss of self-esteem.
Fourthly, the node labeled "searching for perfection" highlights the unrealistic expectations that many women have about the results of cosmetic surgery. The graph suggests that women may have unrealistic expectations about the ability of cosmetic surgery to improve their appearance and boost their self-esteem. This can lead to disappointment and dissatisfaction with the results of surgery, as well as a continued pursuit of perfection through additional surgery.
Finally, the node labeled "stop" highlights the importance of critical thinking and self-reflection when it comes to cosmetic surgery. The graph suggests that women should be aware of the potential negative consequences of cosmetic surgery and should carefully consider the reasons why they are undergoing surgery. They should also be aware of the unrealistic expectations that they may have about the results of surgery and should be prepared to address any negative emotions or feelings of dissatisfaction that may arise.
In conclusion, the explanation graph provides a detailed analysis of the reasons why women undergo cosmetic surgery. The graph highlights the fundamental human desire to feel attractive and confident in one's appearance, the gendered nature of cosmetic surgery, the potential negative consequences of cosmetic surgery, and the importance of critical thinking and self-reflection when it comes to surgery. By understanding these reasons and the potential consequences of cosmetic surgery, women can make informed decisions about whether or not to undergo surgery and can work to address any negative emotions or feelings of dissatisfaction that may arise. |

Table 14: Example of creative chat on `SceneGraphs`. We utilize GPT-4 to generate an image from the graph, assisting readers in visualizing it

| Graph | node_id,node_attr
0,name: head; attribute: black; (x,y,w,h): (386, 185, 8, 4)
1,name: swimming pool; attribute: small, water; (x,y,w,h): (465, 92, 35, 3)
2,name: shore; (x,y,w,h): (49, 85, 436, 108)
3,name: fence; attribute: wired; (x,y,w,h): (11, 158, 167, 46)
4,name: mountains; attribute: covered; (x,y,w,h): (393, 53, 107, 12)
5,name: water; attribute: blue; (x,y,w,h): (0, 50, 499, 154)
6,name: sheep; attribute: walking, grazing; (x,y,w,h): (243, 191, 20, 13)
7,name: sky; attribute: blue, clear, cloudy; (x,y,w,h): (0, 0, 501, 60)
8,name: sheep; attribute: white, fluffy; (x,y,w,h): (409, 176, 13, 10)
9,name: field; attribute: large, green, grassy; (x,y,w,h): (0, 138, 503, 112)
10,name: grass; (x,y,w,h): (0, 84, 500, 166)
11,name: sheep; attribute: white, fluffy; (x,y,w,h): (398, 178, 13, 11)
12,name: sheep; attribute: white, fluffy; (x,y,w,h): (385, 179, 17, 11)
13,name: sheep; attribute: white, fluffy; (x,y,w,h): (364, 181, 14, 10)
14,name: shore; attribute: sandy; (x,y,w,h): (190, 123, 171, 12)
src, edge_attr, dst
0,to the left of,11\n 0,to the right of,13\n 3,to the left of,6\n 6,to the right of,3\n 6,grazing on,10\n 6,in,9\n 6,in,10\n 7,above,5\n 8,eating,10\n 8,to the right of,11\n 8,to the right of,12\n 11,to the left of,8\n 11,to the right of,0\n 11,to the right of,13\n 11,to the right of,12\n 11,eating,10\n 12,eating,10\n 12,to the left of,8\n 12,to the right of,13\n 12,to the left of,11\n 13,to the left of,0\n 13,to the left of,11\n 13,to the left of,12\n 13,eating,10\n |
| --- | --- |
| Image | 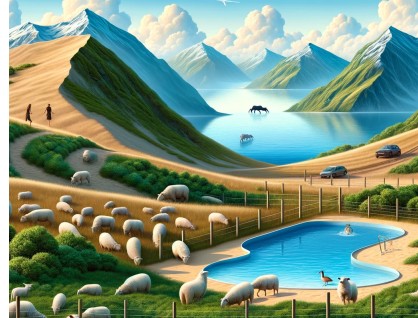 |
| User | Write an 500-word advertisement for this place in the scene graph that would make people want to visit it. |
| Model | Sure! Here's an example of a 500-word advertisement for the place based on the scene graph:
"Experience the breathtaking beauty of our place, where the rolling hills meet the sparkling water of the swimming pool. Take a leisurely stroll through the lush green fields, and let the gentle breeze caress your face as you gaze out at the picturesque view. The sheep graze peacefully in the distance, their white coats glistening in the sunlight. The sky is a brilliant blue, with fluffy white clouds drifting lazily across. Come and bask in the serene ambiance of our place, where the natural beauty will leave you in awe." |

Table 15: Example of creative conversation on the `WebQSP` dataset.

| | |
|---|---|
| Graph | node_id, node_attr
8, benjamin franklin
63, glass harmonica
98, lightning rod
99, stealing god's thunder: benjamin franklin's lightning rod and the invention of america
227, benjamin franklin worked with thomas jefferson
442, franklin stove
525, bifocals
706, benjamin franklin drawing electricity from the sky
1044, prokop diviš

src, edge_attr, dst
706, visual_art.artwork.art_subject, 8
8, base.kwebbase.kwtopic.connections_from, 227
8, book.book_subject.works, 99
525, law.invention.inventor, 8
442, law.invention.inventor, 8
98, law.invention.inventor, 1044
8, law.inventor.inventions, 98
63, law.invention.inventor, 8
8, base.argumentmaps.innovator.original_ideas, 442
98, law.invention.inventor, 8
8, law.inventor.inventions, 525
442, base.argumentmaps.original_idea.innovator, 8
8, law.inventor.inventions, 442
8, law.inventor.inventions, 63 |
| User | Question: what else did ben franklin invent? Surrounding this question, develop an educational tour through the knowledge graph, highlighting interesting facts and connections in an engaging way. |
| Model | Great! Based on the provided knowledge graph, here's an educational tour that highlights interesting facts and connections related to Benjamin Franklin's inventions and other notable achievements:
1. Franklin's Lightning Rod: Start by exploring Franklin's most famous invention - the lightning rod. Discover how Franklin's invention protected buildings from lightning strikes and how it changed the way people understood electricity. Connection: Franklin worked with Thomas Jefferson on the lightning rod, and their collaboration led to the development of the concept of positive and negative charges.
2. Bifocals: Move on to Franklin's invention of bifocal glasses. Learn how Franklin's vision problems inspired him to create a new type of glasses that allowed people to see both near and far objects clearly. Connection: Franklin's bifocal glasses were a groundbreaking invention that improved the lives of millions of people around the world.
3. Franklin Stove: Explore Franklin's design of the Franklin stove, a cast-iron stove that provided more heat with less fuel. Discover how Franklin's stove revolutionized heating systems and how it helped to reduce fuel costs for households and businesses. Connection: Franklin's stove was an early example of energy efficiency and sustainability, and it paved the way for modern heating systems.
4. Prokop Diviš: Learn about Franklin's connection to Prokop Diviš, a Czech inventor who built on Franklin's work and developed the first electric generator. Discover how Diviš's invention helped to power the Industrial Revolution and how it changed the way people lived and worked. Connection: Franklin's work on the lightning rod inspired Diviš to explore the properties of electricity, and their collaborative efforts helped to advance the field of electrical engineering.
5. Innovator and Original Ideas: Explore Franklin's reputation as an innovator and his ability to come up with original ideas. Discover how Franklin's creativity and curiosity led to many of his inventions and how he was able to see connections between seemingly unrelated things. Connection: Franklin's innovative spirit and ability to think outside the box inspired many other inventors and scientists, including Thomas Edison and Nikola Tesla.
6. Collaboration and Networking: Highlight Franklin's collaborative approach to invention and his ability to network with other inventors and scientists. Discover how Franklin's connections with other innovators helped to advance the field of science and technology. Connection: Franklin's collaborations with other inventors and scientists, such as Thomas Jefferson and Prokop Diviš, helped to accelerate the pace of innovation and pave the way for future breakthroughs.
7. Legacy and Impact: Reflect on Franklin's lasting legacy and the impact of his inventions on modern society. Discover how Franklin's inventions continue to inspire new generations of inventors and scientists and how they have helped to shape the world we live in today. Connection: Franklin's inventions and ideas have had a lasting impact on many fields, including science, technology, and society as a whole. His legacy continues to inspire and motivate people around the world.
By following this educational tour through the knowledge graph, students will gain a deeper understanding of Benjamin Franklin's inventions and their impact on modern society. They will also learn about Franklin's collaborative approach to invention, his innovative spirit, and his ability to think outside the box. |

