# OpenReview forum: "G-Retriever: Retrieval-Augmented Generation for Textual Graph Understanding and Question Answering"
_NeurIPS.cc/2024/Conference — NeurIPS 2024 poster_

### Official Review · Reviewer_RHZZ · 2024-07-08

**Soundness:** 3
**Presentation:** 3
**Contribution:** 3
**Rating:** 6
**Confidence:** 4

**Summary:**

This paper introduces G-Retriever, which combines LLMs, GNNs and RAG for graph question-answering tasks. The authors first develop a more comprehensive benchmark named GraphQA. Then they present G-Retriever, which has four main steps including indexing, retrieval, subgraph construction and answer generation. Specifically, G-Retriever use an adapted PCST method, which also consider the importance of the edge semantics. The constructed subgraph and the query are then sent to the LLM for final answer generation. Extensive experiments validate the effectiveness and efficiency of G-Retriever.

**Strengths:**

1. The introduction of the GraphQA benchmark fills a significant gap in the research community by providing a comprehensive benchmark for evaluating graph QA applications.
2. The motivation of the proposed G-Retriever is clear and the paper is well-structured.
3. Extensive experimental results demonstrate that G-Retriever can consistently outperform  baseline models.
4. Codes are provided for reproducibility.

**Weaknesses:**

- In Section 5.1 (Indexing), the authors use a pre-trained LM to encode the text attributes of nodes and edges into representations. However, there is no ablation study to demonstrate the importance of the node attributes or the text attributes, which leaves an unexplored gap in understanding the contribution of these features to the overall performance of the G-Retriever.
- In Section 5.3 (Subgraph Construction), the authors use PCST to obtain a more refined subgraph. However, the motivation for selecting PCST is not clearly explained. Is PCST the optimal for this task? How the "optimally sized and relevant subgraphs" are defined? At least the authors could consider to justify this by comparing some naive approaches, such as fixed number or fixed similarity threshold.
- The claim of the new benchmark is somewhat weak, as it primarily involves the reintroduction of an existing dataset. Additionally, the contributions related to the QA formulation and graph reformatting are limited.
- The evaluation of hallucinations lacks sufficient detail and requires further elaboration.

**Questions:**

Please see the questions in the Weakness section.

**Limitations:**

The authors have discussed the limitations of their work in the paper.

---

> ### Author Rebuttal · Authors · 2024-08-06
>
> We thank the reviewer very much for the careful reading and comments regarding our work. Please, see below our answer to the raised comments/questions.
>
> > **Reviewer:** In Section 5.1 (Indexing), the authors use a pre-trained LM to encode the text attributes of nodes and edges into representations. However, there is no ablation study to demonstrate the importance of the node attributes or the text attributes, which leaves an unexplored gap in understanding the contribution of these features to the overall performance of the G-Retriever.
>
> **Authors:** The text attributes in G-Retriever play a critical role in three main areas: (1) Retrieval: Node and edge embeddings are used to select the top-k nodes and edges based on their semantic similarity to the query embedding. (2) GNN Input: These embeddings are fed into the GNN, which processes the graph structure and refines the embeddings to capture more complex relationships within the graph. (3) LLM Input: The retrieved subgraph is textualized and used as input for the LLM to generate answer.
>
> To address the reviewer’s concern, we have conducted an ablation study to assess the contribution of these features.
>
> | Features | Without Node | Without Edge |
> |----------|--------------|--------------|
> | Retrieval | 66.58 | 58.37 |
> | GNN Input | 68.85 | 67.87 |
> | LLM Input | 56.32 | 68.24 |
>
>
> > **Reviewer:** In Section 5.3 (Subgraph Construction), the authors use PCST to obtain a more refined subgraph. However, the motivation for selecting PCST is not clearly explained. Is PCST the optimal for this task? How the "optimally sized and relevant subgraphs" are defined? At least the authors could consider to justify this by comparing some naive approaches, such as fixed number or fixed similarity threshold.
>
> **Authors:** Due to the character limit, we kindly refer the reviewer to our general response “G2: Elaboration on PCST-Based Retrieval”, where we have addressed this question in detail.
>
> > **Reviewer:** The claim of the new benchmark is somewhat weak, as it primarily involves the reintroduction of an existing dataset. Additionally, the contributions related to the QA formulation and graph reformatting are limited.
>
> **Authors:** Due to the character limit, we kindly refer the reviewer to our general response “G1. Contribution of the GraphQA Benchmark”, where we have addressed this question in detail.
>
>
> > **Reviewers:** The evaluation of hallucinations lacks sufficient detail and requires further elaboration.
>
> **Authors:** Thank you for your feedback. Due to space constraints, the detailed evaluation of hallucinations is provided in Appendix F. Here, we offer a summary:
>
> - **Experiment Design.** We instructed the LLM to answer graph-related questions and list the nodes or edges in the explanation graph that support its answers. Since there are no standard answers, evaluating the LLM’s responses becomes challenging. To address this, we manually examined 100 responses generated by both our method and a baseline method (LLM with graph prompt tuning) to verify whether the referenced nodes and edges actually exist in the graph.
> - **Evaluation Metrics.** We assessed the model’s faithfulness using three metrics: the fraction of valid nodes (Valid Nodes), the fraction of valid edges (Valid Edges), and the fraction of times the entire set of nodes and edges cited was valid (Fully Valid Graphs).
> - **Baseline.** We adapted MiniGPT-4 [57] to graph contexts as our baseline, using a frozen LLM with a trainable GNN that encodes graph data as a soft prompt (LLM+Graph Prompt Tuning). We focused on graph prompt tuning due to the large size of the textual representation of the graph, which often exceeds the input token limits of LLMs.
> - **Results.** As shown in Table 5, G-Retriever outperforms the baseline in reducing hallucinations. The baseline method showed only 31% valid nodes, 12% valid edges, and 8% fully valid node-edge sets. In contrast, G-Retriever achieved 77% validity in nodes, 76% in edges, and 62% overall validity in node-edge sets. These results highlight the effectiveness of G-Retriever in accurately referencing graph elements and significantly reducing hallucinations.
>
> We will ensure that these points are clearly elaborated upon in the revised manuscript for better understanding.

---

> > ### Comment · Reviewer_RHZZ · 2024-08-08
> > **Thank you for your response.**
> >
> > Thank you for your response. I am satisfied with the clarification and will maintain my positive review score.

---

> ### Author Response · Authors · 2024-08-09
> **Response to Reviewer RHZZ**
>
> Thank you very much for your time reviewing our answer.

---

### Official Review · Reviewer_GQPK · 2024-07-12

**Soundness:** 3
**Presentation:** 2
**Contribution:** 2
**Rating:** 5
**Confidence:** 4

**Summary:**

This paper proposes a retrieval-augmented method G-Retriever for graphs with textual attributes. It introduces a graph question answering (GraphQA) benchmark by converting existing graph datasets into uniform format. Subsequently, it proposes a G-Retriever method to answer questions related to the textual graphs. Specifically, it first retrieves top-k most relevant nodes and edges from the textual graphs and then constructs a subgraph based on the retrieved nodes and edges using an existing algorithm called Prize-Collecting Steiner Tree (PCST). Subsequently, G-Retriever leverages an GAT model to obtain the pooled representation of the constructed subgraph, which is used as a soft prompt for an LLM to generate the answer.

**Strengths:**

The idea of retrieving subgraphs from textual graphs to augment the LLM is interesting. Experimental results on three datasets in the GraphQA benchmark demonstrate the effectiveness of the proposed G-Retriever model. The code is provided for reproducibility.

**Weaknesses:**

1. The paper mentions “chat with their graph” in the abstract and the introduction. However, it is not very clear what this concept means. Additionally, after the abstract and introduction, there are no further introductions or explanations about this concept.

2. In line 69, the paper states that the proposed G-Retriever can improve the explainability. However, there are no empirical analyses regarding the explainability of the proposed method.

3. The novelty of the proposed GraphQA benchmark seems limited, as it only converts three existing graph datasets to a uniform format.

4. I think the main novelty of the proposed G-Retriever model is that it leverages a PCST algorithm to find a connected subgraph between the retrieved nodes and edges. However, the paper does not clearly explain the advantages of retrieving a subgraph. Why is it necessary to retrieve a subgraph rather than directly using the texts of the retrieved nodes and edges as inputs for the LLM? Constructing a subgraph could potentially introduce noises, which might negatively impact the performance. Additionally, the paper does not adequately justify the use of PCST for finding the subgraph. Have the authors explored other alternative methods, such as the shortest paths between the retrieved nodes?

5. There are no quantitative or qualitative analyses regarding the quality of the retrieved subgraphs. Providing such analyses could offer some insights into the effectiveness of the subgraphs.

6. In the efficiency evaluation (section 6.3), the paper only compares the training times of G-Retriever and its variant without retrieval (I assume this is what “Before Retrieval” means, as there are no explanations about the columns in Table 4). However, it would be more appropriate to compare the inference times of G-Retriever and its variant during the inference stage. G-Retriever requires additional steps of retrieving nodes and edges, as well as constructing subgraphs using the PCST algorithm, which may result in longer inference times compared to not using retrieval.

7. A lot of necessary experimental details in section 6.4 and 6.5 are provided in the Appendix, which may hinder the readers’ understanding of the results and analyses in these sections. For instance, it is impossible to understand what “Valid Nodes”, “Valid Edges” and “Fully Valid Graphs” mean in Table 5 without referring to the Appendix. Additionally, in section 6.4, the paper uses manual examination to evaluate the hallucination performance of LLMs. However, it is unclear how many annotators were involved in this process and whether there is a potential for biases during the evaluation.

8. The texts in Figure 1 and Figure 2 are too small to read.

**Questions:**

1. What does “chat with their graph” mean and how can the proposed G-Retriever model achieve this purpose?

2. How can the proposed G-Retriever improve the explainability? Are there any qualitative analyses to support this argument?

3. What are the advantages of using the PCST algorithm to retrieve subgraphs?

4. How to evaluate the quality of the retrieved subgraphs?

**Limitations:**

The authors have discussed the limitations of their work in the paper.

---

> ### Author Rebuttal · Authors · 2024-08-06
>
> We wish to thank the reviewer very much for the careful reading and comments regarding our work. Please, see below our answer to the raised comments/questions.
>
> > **Reviewer (W1 & Q1):** Explanation on “chat with their graph”
>
> **Authors:** By “chat with their graph,” we mean that users can interact with the graph through a conversational interface. Users can input a textual graph and pose natural language queries, to which G-Retriever will respond in natural language. For example, as shown in Figure 1, if a user provides a mindmap-like explanation graph and requests an argument essay, G-Retriever can generate the essay accordingly. This feature enhances human-AI interaction, making the model more intuitive and aligned with human expectations. We will clarify and elaborate on this concept in the main sections of our revised manuscript.
>
> > **Reviewer (W2 & Q2):** Explainability of G-Retriever
>
> **Authors:** We believe G-Retriever enhances explainability in the following ways:
> - **Retrieved subgraph.** By returning the most relevant subgraph in response to a query, users can see which parts of the graph are considered important for the answer. This helps users understand the basis of the model’s responses. For example, if users want to understand why certain information is present or absent in the LLM’s response, they can inspect the subgraph to see whether such information is present or absent in the retrieved subgraph.
> - **Conversational Interface.** G-Retriever allows users to ask follow-up questions and receive detailed natural language explanations. For example, if a user questions the LLM’s response, they can ask, “Why do you think [xxx]? Please explain your answer.” This interactive capability enables users to explore the model’s reasoning process and gain deeper insights into how it interprets graph data.
>
> We will include specific examples in the revised version of our manuscript to illustrate the two explainability properties mentioned above.
>
> > **Reviewer (W3):** Novelty of GraphQA benchmark
>
> **Authors:** Due to the character limit, we kindly refer the reviewer to our general response “G1: Contribution of the GraphQA Benchmark”, where we have addressed this question in detail.
>
> > **Reviewer (W4 & Q3):** PCST and alternative retrieval methods
>
> **Authors:** Due to the character limit, we kindly refer the reviewer to our general response “G2: Elaboration on PCST-Based Retrieval”, where we have responded to this question in detail.
>
> > **Retriever (W5 & Q4):** Quality of the retrieved subgraphs
>
> **Authors:** We quantify the quality of our retrieval method as follows:
>
> We examine the retrieval subgraph; if the label is contained within it, we consider it a successful retrieval. We calculate the retrieval success rate of our method and the retrieval method proposed in KAPING [1] on the WebQSP dataset. The results are as follows:
> - Hit@1 accuracy for [KAPING with top-k triple retrieval] is 60.81%.
> - Hit@1 accuracy for [Ours with PCST-based subgraph retrieval] is 67.87%.
>
> This demonstrates the effectiveness of our method.
>
>
> > **Reviewer (W6):** Efficiency evaluation
>
> **Authors:** We conducted additional experiments on the WebQSP dataset comparing inference times of G-Retriever with and without retrieval:
> | Method                       | Time in minutes | Accuray with Hit@1  |
> |------------------------------|----------|--------|
> | G-Retriever                  | 9.55     | 70.49  |
> | G-Retriever (w/o retrieval)  | 21.01    | 63.84  |
>
>
> Despite additional steps, G-Retriever is faster (9.55 vs. 21.01 minutes) and more accurate (70.49% vs. 63.84%). The speedup is due to:
>
> - PCST subgraph construction is very efficient. For instance, in the WebQSP dataset, even the largest subgraph (2,000 nodes, 6,104 edges) takes only 0.29 seconds to construct.
> - PCST retrieval reduces the graph size significantly, eliminating up to 99% of nodes in the WebQSP dataset, which in turn speeds up inference time.
>
>
>
> > **Reviewer (W7):** Experimental details
>
> **Authors:** In Table 5, we assessed the model’s faithfulness using three metrics: the fraction of valid nodes (denoted as Valid Nodes), the fraction of valid edges (denoted as Valid Edges), and the fraction of times the entire set of nodes and edges cited was valid (denoted as Fully Valid Graphs). As recommended, we will update the table captions in our revised manuscript to include this information.
>
> Regarding the hallucination evaluation in section 6.4, the manual examination was conducted by the authors. We will develop the following points:
>
> **1. Necessity of Manual Evaluation:** Since standard answers are not available for these questions, the LLM’s responses are inherently flexible. This flexibility, along with the varied output formatting, makes automatic evaluation difficult, necessitating manual review.
>
> **2. Objectivity in Determining Hallucinations:** The determination of hallucinations is based on objective criteria. For example, in Table 1, in the response from “LLM w/ Graph Prompt Tuning,” it is stated: “ The animal in the bushes is a deer. Nodes: * Deer (node 1), * Bushes (node 2); Edges: * Deer → Bushes (edge 1) * Deer → Grass (edge 2) * Bushes → Grass (edge 3)”. The fraction of “valid nodes” is 1/2 = 0.5, as “Bushes” exists in the scene graph while “Deer” does not. The fraction of “valid edges” is 0/3 = 0, since none of the mentioned edges exist in the scene graph. As there are hallucinated nodes and edges, this is not a “fully valid graph.”
>
> **3. Transparency and Bias Mitigation:** We recognize the importance of transparency and minimizing bias in our evaluation process. To address potential concerns, we will open-source the model outputs used for these manual checks, allowing for public scrutiny and verification.
>
> > **Reviewer (W8):** The texts in Figure 1 and Figure 2 are too small to read.
>
> **Authors:** We will enlarge the text in Figures 1 and Figure 2 to enhance readability in our next manuscript.

---

> > ### Comment · Reviewer_GQPK · 2024-08-08
> >
> > I would like to thank the authors for their responses, which address many of my concerns. I am satisfied with the clarifications regarding the PCST algorithm, the analyses of retrieved subgraphs and the efficiency.
> >
> > However, as noted by the authors, the paper could be further improved by providing more details in the following areas: (1) "chat with their graph" concept: further explanation and examples are needed; (2) Explainability of G-Retriever: empirical analyses to support this claim should be included; (3) Experimental Details: necessary experimental details should be provided in the main text to facilitate the understanding of the results.
> >
> > Therefore, I have updated my score accordingly.

---

> > > ### Author Response · Authors · 2024-08-09
> > > **Response to Reviewer GQPK**
> > >
> > > Thank you very much for your time reviewing our answer, and for updating your score. We appreciate your suggestions and will ensure to address the areas you've highlighted, including further details on the "chat with their graph" concept, explainability analyses, and necessary experimental details in the revised manuscript.

---

### Official Review · Reviewer_mCtm · 2024-07-12

**Soundness:** 4
**Presentation:** 3
**Contribution:** 4
**Rating:** 7
**Confidence:** 4

**Summary:**

This paper proposed a Graph Question Answering (GraphQA) benchmark with data collected from different tasks including ExplaGraphs, SceneGraphs, and WebQSP. Then, they proposed G-Retriever method, introducing the first retrieval-augmented generation (RAG) approach for general textual graphs, which can be fine-tuned to enhance graph understanding via soft prompting. G-Retriever performs RAG over a graph by formulating this task as a Prize-Collecting Steiner Tree optimization problem. The generation model is fine-tuned by a graph token and the textualized graph. Empirical evaluations show G-Retriever outperforms baselines on textual graph tasks from multiple domains, scales well with larger graph sizes.

**Strengths:**

- This proposed GraphQA benchmark is comprehensive and comes in a timely manner.
- Converting the problem of finding a connected subgraph that maximizes the total prize values of its nodes while minimizing the total costs of its edges to PCST is smart.
- Strong experimental results by using the proposed G-retriever on all three datasets.

**Weaknesses:**

I don't see major weaknesses of this paper. Several minor points:
- I think several baselines are missing: (1) Simply feed the retrieved top-k nodes/edges (plus its neighbors) to the LLM would be able to show the effectiveness of PCST. (2) It would be helpful to understand the generation model by replacing the tuned generation model to a fixed LLM like GPT4, Claude, etc.
- When encoding the nodes and edges, the context of the nodes/edges is missing, it could be beneficial to include the context (e.g., adjacent nodes)
- Table captions should be more informative so that they are easier to understand (e.g., In Table5, what metrics are used?)

**Questions:**

n/a

---

> ### Author Rebuttal · Authors · 2024-08-06
>
> We would like thank the reviewer very much for the careful reading and comments regarding our work. Please, see below our answer to the raised comments/questions.
>
> > **Reviewer:** I think several baselines are missing: (1) Simply feed the retrieved top-k nodes/edges (plus its neighbors) to the LLM would be able to show the effectiveness of PCST.
>
> **Authors:** To demonstrate the effectiveness of PCST, we compared it to a similar baseline, KAPING [1]. KAPING retrieves the top-k triples related to the question from the graph, adds them to the input question as a prompt, and then sends this to LLMs to generate the answer.
> In addition to KAPING, we included the following baseline methods to address the reviewer's concern:
> - Top-k nodes plus their neighbors: For each query, For each query, the top-k nodes and their one-hop neighbors are retrieved.
> - Shortest path retrieval: This approach involves retrieving the top-k nodes and the shortest paths between them.
> For all methods, we set k = 5 and used llama2-7b-chat as the LLM. The results are presented in the table below, where we observed that our PCST-based retrieval outperforms the baseline retrieval methods, achieving an accuracy of 66.17 on the WebQSP dataset.
>
> | Method                               | Hit@1 |
> |--------------------------------------|--------------------|
> | PCST retrieval                        | 66.17              |
> | top-k triples retrieval (KAPING)      | 52.64              |
> | top-k nodes plus its neighbors        | 49.82              |
> | shortest path retrieval               | 55.20              |
>
> > **Reviewer:** (2) It would be helpful to understand the generation model by replacing the tuned generation model to a fixed LLM like GPT4, Claude, etc.
>
> **Authors:** To address this concern, we conducted additional experiments by replacing the tuned generation model with fixed LLMs on the WebQSP dataset. Specifically, we first applied the PCST retrieval to obtain the subgraph, then converted it into text and fed it into the fixed LLMs along with the query. We considered two fixed LLMs: the open-source llama2-7b-chat and the closed-source GPT-4. The results are summarized in the table below:
> |  Method                   | Hit@1 |
> |---------------------|--------------------|
> | llama2-7b-chat      | 66.17              |
> | GPT-4o              | 67.87              |
> | G-Retriever | 70.49              |
>
> As the results indicate, G-Retriever outperforms both fixed LLM baselines. This demonstrates the effectiveness of our approach in the tuned generation model.
>
> > **Reviewer:** When encoding the nodes and edges, the context of the nodes/edges is missing, it could be beneficial to include the context (e.g., adjacent nodes)
>
> **Authors:** Thank you for your observation. In our indexing step, we use a pre-trained language model (LM) to encode the text attributes associated with each node and edge into embeddings. Although we do not explicitly include the context, such as adjacent nodes, during this initial encoding, we address this in the generation step. Specifically, our framework incorporates a graph neural network (GNN) component designed to aggregate information from adjacent nodes and update the node and edge representations based on the graph context. This approach allows us to effectively capture and use the context during the generation phase. Therefore, while the context is not directly included in the initial encoding, it is integrated later in the process through the GNN, ensuring that the contextual information is considered in the overall framework.
>
> > **Reviewer:** Table captions should be more informative so that they are easier to understand (e.g., In Table5, what metrics are used?)
>
> **Authors:** Thank you for the suggestion. In Table 5, we assessed the model’s faithfulness using three metrics: the fraction of valid nodes (denoted as Valid Nodes), the fraction of valid edges (denoted as Valid Edges), and the fraction of times the entire set of nodes and edges cited was valid (denoted as Fully Valid Graphs). As recommended, we will update the table captions in our revised manuscript to include this information.
>
>
>
> **Reference**
>
> [1] Knowledge-Augmented Language Model Prompting for Zero-Shot Knowledge Graph Question Answering, 2023.

---

> > ### Comment · Reviewer_mCtm · 2024-08-08
> > **Thanks for the response**
> >
> > Thank you for your response. I think adding those comparison does make the argument more solid and I will maintain my rating.

---

> > > ### Author Response · Authors · 2024-08-09
> > > **Response to Reviewer mCtm**
> > >
> > > Thank you very much for your time reviewing our answer. We will ensure to include these comparisons in our revised manuscript.

---

### Official Review · Reviewer_GHV4 · 2024-07-13

**Soundness:** 3
**Presentation:** 3
**Contribution:** 3
**Rating:** 7
**Confidence:** 4

**Summary:**

The work builds a new GraphQA benchmark for real-world graph question answering and presents G-Retriever, an architecture adept at complex and creative queries. Given a query and a graph, G-Retriever retrieves a connected subgraph from the original graph according to the query. The subgraph,  a textualized graph, and the query are fed into an architecture consisting of a graph encoder aligned with an LLM. Experiments show that G-Retriever achieves SoTA in textual graph tasks across multiple domains, significantly improves efficiency, and demonstrates resistance to hallucination.

**Strengths:**

1. The idea of incorporating retrieval for graph question answering is novel and interesting and could be inspiring for future works.
2. The proposed G-retriever framework performs well on the GraphQA dataset. The retrieval method scales effectively with larger graph sizes, which is not addressed in many previous works. Locating the related subgraph instead of using the entire graph also reduces hallucination.
3. The experiments are comprehensive, including analysis from various aspects such as hallucination and graph encoder selection.

**Weaknesses:**

1. The framework seems to mainly work for larger graphs. For tasks with smaller graphs, such as the ExplaGraphs dataset with an average node number of  5.17, it is unnecessary to use such a framework. The performances of G-retrieval and GraphToken are nearly the same for the ExplaGraphs dataset.

**Questions:**

1. After the subgraph is retrieved, is it possible to use neuro-symbolic methods (e.g. answer set programming or graph search) instead of using a graph encoder together with an LLM? It would be more efficient and cost much less.

**Limitations:**

Yes

---

> ### Author Rebuttal · Authors · 2024-08-06
>
> We thank the reviewer very much for the careful reading and comments regarding our work. Please, see below our answer to the raised comments/questions.
>
> > **Reviewer:** The framework seems to mainly work for larger graphs. For tasks with smaller graphs, such as the ExplaGraphs dataset with an average node number of 5.17, it is unnecessary to use such a framework. The performances of G-retrieval and GraphToken are nearly the same for the ExplaGraphs dataset.
>
> **Authors:** We acknowledge that G-Retriever and GraphToken demonstrate similar performance on the ExplaGraphs dataset, which consists of small graphs. As noted by the reviewer, other techniques like GraphToken can be employed effectively on this dataset. However, our aim in applying G-Retriever to ExplaGraphs dataset was to highlight the flexibility and effectiveness of the proposed technique across a wide range of graph sizes. This versatility is particularly advantageous in datasets like SceneGraphs, where graph sizes, although small (averaging 19.1 nodes) and varying significantly (ranging from 1 to 126 nodes with a standard deviation of 8.3). For these small-scale graphs, G-Retriever significantly outperforms GraphToken, achieving 0.8131 accuracy compared to 0.4903, as reported in Table 5. Additionally, G-Retriever offers extra benefits, such as a flexible conversational interface. As shown in Figure 1, G-Retriever can generate an essay based on a given explanation graph. We believe that this versatile question-answering capability adds substantial value beyond just accuracy.
>
>
> |     | # node: min, max, mean ± std | # edge: min, max, mean ± std|
> | -------- | ------- | ------- |
> | ExplaGraphs  | 4, 9, 5.17 ± 1.18    | 3, 8, 4.25 ± 1.23 |
> | SceneGraphs | 1, 126, 19.13 ± 8.28     | 0, 1657, 68.44 ± 65.77 |
> | WebQSP    | 3, 2000, 1371.18 ± 566.91    | 2, 10818, 4253.27 ± 2238.12 |
>
>
>
> > **Reviewer:** After the subgraph is retrieved, is it possible to use neuro-symbolic methods (e.g. answer set programming or graph search) instead of using a graph encoder together with an LLM? It would be more efficient and cost much less.
>
> **Authors:** Thank you for the insightful suggestion. While neuro-symbolic methods such as answer set programming or graph search can indeed offer efficiency and cost-effectiveness, they generally require a predefined specification of patterns, i.e. symbols, to guide the search process. In our graph QA scenario, the patterns are highly dependent on the query, which can vary significantly in complexity, making it challenging to effectively predefine these patterns. However, we acknowledge that for specific applications where patterns can be clearly defined, neuro-symbolic approaches could be a viable alternative and are certainly worth exploring in future work.

---

> > ### Comment · Reviewer_GHV4 · 2024-08-09
> > **Thanks for your response**
> >
> > Thanks for the response. My concerns are well addressed. Overall, I think this work makes a good contribution to the area. I will improve my score to 7.

---

> > > ### Author Response · Authors · 2024-08-09
> > > **Response to Reviewer GHV4**
> > >
> > > Thank you very much for your time reviewing our answer, and for updating your score.

---

### Author Rebuttal · Authors · 2024-08-06

We sincerely thank the reviewers for their time and effort in evaluating our paper.

In this global response, we aim to clarify the contributions of the GraphQA benchmark (G1), elaborate on our PCST-based retrieval method (G2), and present new experiments conducted in response to the reviewers' feedback (G3).

**G1. Contribution of the GraphQA Benchmark:**
We acknowledge that the GraphQA benchmark involves converting three existing graph datasets into a uniform format. However, we believe this standardization provides significant value to the research community in several ways:

- **Task Introduction:** Unlike existing graph question-answering benchmarks that focus on small or synthetic graphs, our benchmark includes real-world applications and frames them as graph question-answering tasks.
- **Standardization:** A key and significant effort of this benchmark is the standardization and processing of diverse datasets into a uniform format suitable for GraphQA tasks. These datasets, previously used in different contexts, are redesigned to focus specifically on GraphQA, ensuring consistent and comparable evaluations across models.
- **Accessibility:** We have open-sourced the GraphQA benchmark, providing a unified format that simplifies model application across multiple datasets. This reduces the complexity of handling various data structures and preprocessing pipelines, lowering barriers for new researchers and encouraging broader participation. We have already seen several novel works using our GraphQA benchmark, and we expect rapid adoption within the LLM and GNN communities.
- **Baseline Comparisons:** The benchmark offers baseline performance metrics, helping researchers identify the strengths and weaknesses of new approaches compared to established baselines.

We will ensure these contributions are more clearly highlighted in the revised manuscript.

**G2. Elaboration on PCST-Based Retrieval:**

**1) Modeling motivation.** We formulate subgraph retrieval as a Prize-Collecting Steiner Tree (PCST) optimization problem. This is motivated by the need to find a connected subgraph containing most relevant nodes and edges, a goal that aligns well with the objectives of PCST: maximizing node values while minimizing edge costs. Though not universally acknowledged as optimal, we have empirically validated its effectiveness.

**2) Effectiveness Compared to Other Retrieval Baselines.** To further demonstrate the effectiveness of PCST-based retrieval, we compared it to a similar baseline, KAPING [1]. KAPING retrieves the top-k triples related to the question from the graph, adds them to the input question as a prompt, and then sends this to LLMs to generate the answer. In addition to KAPING, we included the following baseline methods to address the reviewers' concern:

- Top-k nodes plus their neighbors: For each query, For each query, the top-k nodes and their one-hop neighbors are retrieved.
- Shortest path retrieval: This approach involves retrieving the top-k nodes and the shortest paths between them.

For all methods, we set k = 5 and used llama2-7b-chat as the LLM. The results are presented in the table below, where we observed that our PCST-based retrieval outperforms the baseline retrieval methods, achieving an accuracy of 66.17 on the WebQSP dataset.
| Method                               | Hit@1 |
|--------------------------------------|--------------------|
| PCST retrieval                        | 66.17              |
| top-k triples retrieval (KAPING)      | 52.64              |
| top-k nodes plus its neighbors        | 49.82              |
| shortest path retrieval               | 55.20              |



**3) Advantages of Subgraph-Based Retrieval.**
- **Context-Relevant.** Selecting nodes and edges in isolation may overlook neighborhood information. In contrast, PCST-based retrieval is guaranteed to return a connected subgraph, capturing the graph context during the retrieval process. This approach retrieves not only high-relevance nodes or edges but also “bridge” elements that connect these with contextually significant nodes or edges, which are crucial for generating a comprehensive response.
- **Size Management.** Compared to the shortest path method, PCST retrieval provides greater control over the size of the retrieved subgraph. By adjusting the prizes and costs on nodes and edges, users can fine-tune the subgraph's extent. In contrast, the shortest path approach lacks the ability to control the distance between the top-k nodes, which can lead to disconnected subgraphs or the inclusion of unnecessarily long paths.


**G3. New experiments:**

In response to reviewers' feedback, we conducted several new experiments to address their concerns:

- Comparison with more retrieval baselines, demonstrating the effectiveness of our PCST-based retrieval method (Table 16).
- Replacing the tuned generation model in G-Retriever with a fixed LLM (Table 17), highlighting the necessity of tuning the generation model, particularly through graph soft prompting in G-Retriever.
- Efficiency evaluation during the inference stage (Table 18), showing the significant efficiency gains brought by the retrieval process.
- Quantifying the quality of our retrieval method (Table 19) shows that our approach retrieves more accurate information than the KAPING baseline.
- Ablation study to demonstrate the importance of node and edge attributes (Table 20).

Please, find the new experiments in the newly rebuttal single-page PDF file.

We hope that our point-to-point responses with the new experiments have clarified the reviewers’ concerns. We are happy to answer any additional questions to clarify further.

Thank you again for your dedicated effort and time to review the paper.

Best regards,

The authors



**References**

[1] Knowledge-Augmented Language Model Prompting for Zero-Shot Knowledge Graph Question Answering, 2023.

---

### Decision · Program_Chairs · 2024-09-25

**Decision:**

Accept (poster)

**Comment:**

This paper proposes a new method named G-Retriever, which is a retrieval-augmented generation (RAG) approach for general textual graphs, and develops a Graph Question Answering (GraphQA) benchmark with data collected from different tasks, such as scene graph understanding, common sense reasoning, and knowledge graph reasoning. The reviewers in general agree that the idea is novel and interesting, the experimental results are strong, and the compiled GraphQA benchmark is useful for the research community. There are some questions raised during the reviews which seem to have been addressed after rebuttal. The authors may revise the paper based on the discussions in the rebuttal.